# Prospects and challenges for squeezing-enhanced optical atomic clocks

Marius Schulte [1✉], Christian Lisdat [2], Piet O. Schmidt [2,3], Uwe Sterr [2] & Klemens Hammerer [1✉]

Optical atomic clocks are a driving force for precision measurements due to the high accuracy and stability demonstrated in recent years. While further improvements to the stability have been envisioned by using entangled atoms, squeezing the quantum mechanical projection noise, evaluating the overall gain must incorporate essential features of an atomic clock. Here, we investigate the benefits of spin squeezed states for clocks operated with typical Brownian frequency noise-limited laser sources. Based on an analytic model of the closed servo-loop of an optical atomic clock, we report here quantitative predictions on the optimal clock stability for a given dead time and laser noise. Our analytic predictions are in good agreement with numerical simulations of the closed servo-loop. We find that for usual cyclic Ramsey interrogation of single atomic ensembles with dead time, even with the current most stable lasers spin squeezing can only improve the clock stability for ensembles below a critical atom number of about one thousand in an optical Sr lattice clock. Even with a future improvement of the laser performance by one order of magnitude the critical atom number still remains below 100,000. In contrast, clocks based on smaller, non-scalable ensembles, such as ion clocks, can already benefit from squeezed states with current clock lasers.

[1] Institute for Theoretical Physics and Institute for Gravitational Physics (Albert-Einstein-Institute), Leibniz University Hannover, Appelstrasse 2, 30167 Hannover, Germany. [2] Physikalisch-Technische Bundesanstalt (PTB), Bundesallee 100, 38116 Braunschweig, Germany. [3] Institute for Quantum Optics, Leibniz University Hannover, Welfengarten 1, 30167 Hannover, Germany. ✉email: marius.schulte@itp.uni-hannover.de; klemens.hammerer@itp.uni-hannover.de

n recent years, atomic clocks based on optical transitions[1] have achieved unprecedented levels in accuracy and stability as frequency references[2–5]. Apart from a redefinition of the SI second, this also facilitates tests of physics beyond the Standard Model[6–9] and opens up the field of relativistic geodesy[10–12]. For these applications, high clock stability is vital in order to reach a given frequency uncertainty in the shortest possible time. Accordingly, approaches from quantum metrology[13] are being pursued which promise to achieve an improvement through the use of entangled atoms. In particular, spin squeezed states[14–16] received much attention due to their practicability and noise resilience[13,17]. Spin squeezed states can be generated with trapped ions[18,19] and in cold atomic gases[20–22], and have already been used in proof-of-principle experiments to demonstrate a reduction of quantum projection noise (QPN) in measurements of small phases on microwave transitions[23–26]. The realization of such tailored entangled states on optical clock transitions is a major challenge for experiment[26–28] and theory[29–34].

In view of these advances, it is important to note that under practical conditions, optical atomic clocks are not exclusively limited by QPN. Indeed, the operating point of a clock at which maximum stability is achieved is determined by a balance of QPN and other noise processes, such as laser phase noise and dead time effects[35–38]. While the instability due to dead time can be considered a merely technical problem, we emphasize that laser phase noise must not be treated as such. Indeed, the suppression of laser noise (fundamentally limited by thermal noise[39] or quantum noise[40]) by locking on an atomic reference is the central objective of an optical atomic clock. To dismiss this noise as a technical imperfection would render the problem trivial. On the other hand, atomic spontaneous decay can be neglected for the most advanced clocks which employ clock transitions with upper state lifetimes way beyond the laser coherence times[1]. We assess the prospects and limitations for improving the stability of optical atomic clocks using spin squeezing under these conditions. We stress that the derived limitations apply to single atomic clocks with conventional (Ramsey) interrogation sequences with squeezed input states. The limitations could be avoided with schemes achieving dead-time-free interrogation or overcoming

laser phase noise[41–47]. The potential gain from entanglement should then be assessed by an appropriate analysis, incorporating the tradeoffs discussed here.

In this work we show that at a given level of dead time and laser phase noise, spin squeezing can only offer an advantage for atomic ensembles below a certain critical number of clock atoms. For state-of-the-art high-quality clock lasers, this critical atomic number is smaller than the size that can realistically be reached in optical lattice clocks without being limited by density effects. Thus, in lattice clocks spin squeezing can only provide an advantage with significant improvements in dead time and phase noise of next generation clock lasers. In contrast, in atomic clocks based on platforms whose atomic number cannot be easily scaled, such as multi-ion traps[48–51] or tweezer arrays[52–55], spin squeezing can offer a relevant advantage.

## Results

**Setup and clock stability.** In optical atomic clocks a laser of high but finite coherence time is stabilized by a control loop to an atomic transition of frequency $\nu_0$, see Fig. 1a. The laser frequency is compared to the atomic transition in a sequence of interrogation cycles, each of duration $T_C$. We consider here Ramsey interrogations with interrogation time $T_R$, and cycles with a dead time $T_D = T_C - T_R$, see Fig. 1b. At the end of an interrogation cycle, the collective atomic spin is measured along a projection, which we take as $S_y$, providing information about the deviation of the laser from the atomic transition frequency, see Fig. 1c. The measurement result is converted into an error signal that is used to correct the laser frequency. The clock instability achieved in this way after averaging over a time $\tau \gg T_C$ is measured in terms of the Allan deviation $\sigma_y(\tau)$ for fractional frequency fluctuations[1]. Later on, we will typically refer to Allan deviations at $\tau = 1$ s only. What is meant by this is that we look for the pre-factor to the asymptotic $\sigma_y(\tau) \propto 1/\sqrt{\tau}$ scaling, found e.g., by extrapolating the Allan deviation from a regime with $\tau \gg T_C$ back to $\tau = 1$ s. Even though the actual stability of the clock at $\tau = 1$ s may be different, e.g., due to the transient response of the feedback loop, this quantity still provides us with a useful measure to compare the

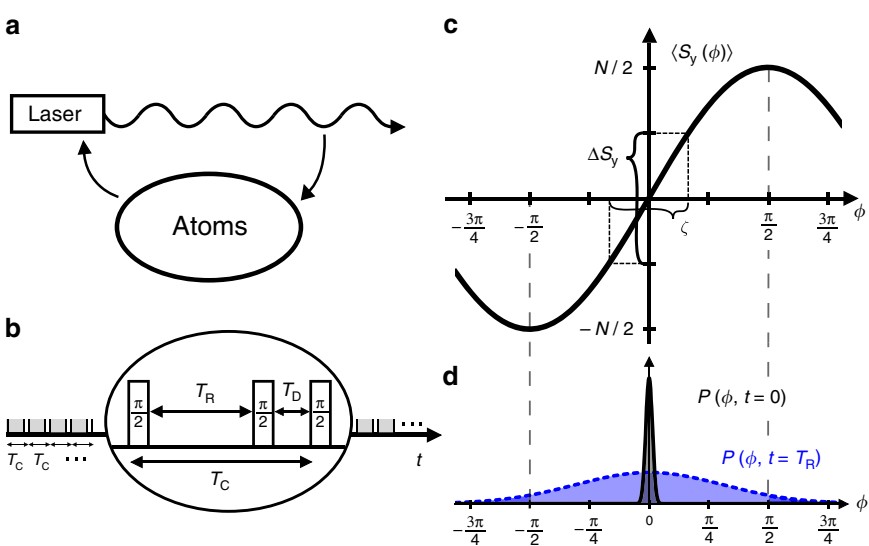

**Fig. 1 Setup and noise processes. a** Measurement and feedback loop to stabilize the laser frequency to an atomic transition. **b** Periodic measurements with Ramsey time $T_R$ and dead time $T_D$ in each cycle of total time $T_C$ lead to increased instability from the Dick effect. **c** Quantum projection noise $\Delta S_y$ for $N$ particles limits the clock stability for short interrogation times but can be decreased with squeezed states thus reducing the inferred phase uncertainty $\zeta = \xi/\sqrt{N}$ where $\xi$ is the Wineland spin squeezing parameter. **d** For longer $T_R$ the distribution $P(\phi, t)$ of phases broadens substantially due the laser's decoherence. Inefficient feedback for phases outside the $[-\frac{\pi}{2}, \frac{\pi}{2}]$ interval gives the coherence time limit.

long term stability of different clocks without limitations based on their specific mode of operation.

For an atomic clock whose stability is exclusively limited by the QPN of the spin measurements $\Delta S_y$, the Allan deviation would asymptotically be[14]

$$\sigma_{\text{QPN}}(\tau) = \frac{1}{2\pi\nu_0 T_R}\sqrt{\frac{T_C}{\tau}}\frac{\xi}{\sqrt{N}}. \qquad (1)$$

Here, $N$ is the number of clock atoms and $\xi = \sqrt{N}\Delta S_y/\langle S_x\rangle$ is the Wineland spin squeezing parameter[14]. For uncorrelated atoms in a coherent spin state with mean spin polarization along $\langle S_x\rangle$, where $\xi = 1$, the QPN scales as $1/\sqrt{N}$, the standard quantum limit. Correlated states of atoms with $\xi < 1$ can optimally change this scaling up to $1/N$[13]. In particular spin squeezed states can reduce the QPN while maintaining a strong spin polarization, thus lowering $\xi$ and ultimately $\sigma_{\text{QPN}}$.

As Eq. (1) suggests, the stability can also be improved by increasing the interrogation time $T_R$, provided the QPN still remains the dominant noise process. Obviously, it will be beneficial to increase $T_R$ to a point where this is no longer the case, and the QPN is reduced to a level where other processes contributing to the clock instability become comparable. Which other noise processes become relevant first depends on the type of atomic clock. For the narrow-band transitions that can be used in optical atomic clocks it is the finite coherence time of the clock laser rather than that of the atoms that is the limiting factor. Laser phase noise affects clock stability in two ways: Firstly, by phase diffusion during dead time (see Fig. 1b), the so-called Dick effect[56] whose contribution to the Allan deviation $\sigma_{\text{Dick}}$ is well known and summarized below. Second, by phase diffusion during the interrogation, causing the distribution of the phase prior to the measurement to become wider. When the Ramsey dark time $T_R$ becomes comparable to the laser coherence time, the differential phase noise between laser and atomic reference can exceed the invertible domain of the Ramsey signal and thus no unambiguous estimate based on the measurement result is possible, as illustrated in Fig. 1d. At this point, the feedback loop becomes ineffective, compromising stability in two ways: First, the finite laser coherence time contributes to the Allan deviation in the form of an additional diffusion process, which we refer to in the following as the laser coherence time limit (CTL). Building on previous work by Leroux et al.[36] and André et al.[57,58], we

develop below a detailed stochastic model of the CTL from which we can infer its contribution to the Allan deviation $\sigma_{\text{CTL}}$. Second, laser phase noise can also result in an abrupt loss of clock stability when the stabilization passes to an adjacent fringe, causing the clock to run permanently wrong. We will show that the resulting limitation of the Ramsey time can be understood quantitatively in the framework of our stochastic model as a first escape time, giving good agreement with previous phenomenological estimates[36]. We find that in the regime of a good atomic clock (long laser coherence time and small dead time) fringe-hops and the CTL contribute either at a similar level or the diffusive process $\sigma_{\text{CTL}}$ constitutes the more stringent limitation for the Ramsey interrogation, so that we concentrate the discussion on the latter effect.

Incorporating these additional effects, the optimal operating point of the control loop has to be determined from a tradeoff between QPN, Dick effect, and CTL, by minimizing the combined instability

$$\sigma_y(\tau) = \sqrt{\sigma_{\text{QPN}}^2(\tau) + \sigma_{\text{Dick}}^2(\tau) + \sigma_{\text{CTL}}^2(\tau)}. \qquad (2)$$

**Lower bound to the combined instability**. Without already going into the specific functional dependence of $\sigma_{\text{Dick}}$ and $\sigma_{\text{CTL}}$ on the parameters that characterize the atomic clock, we can highlight the most important features, most of which are intuitive to understand: Just as the QPN, the Dick noise is monotonically decreasing with longer Ramsey time as the relative weight of the dead time $T_D$ goes down. However the CTL will increase with $T_R$, as explained above. In contrast to QPN, both Dick and CTL noise do not depend on the size of the atomic ensemble $N$. This should be clear for the Dick effect, which is determined by the laser noise, $T_D$ and $T_R$ only. The fact that the CTL does not depend on $N$ is not so obvious, and will be shown below. These scalings are visible in Fig. 2a which shows the combined Allan deviation, Eq. (2), and all three contributing noise processes versus Ramsey time for a small ensemble ($N = 10$, blue solid line) and a larger ensemble of atoms ($N = 2000$, red solid line) in a coherent spin state. Solid lines in Fig. 2a correspond to the analytical models, symbols show the results of numerical simulations of the closed feedback loop (see supplementary note 1) in excellent agreement with the theoretical curves.

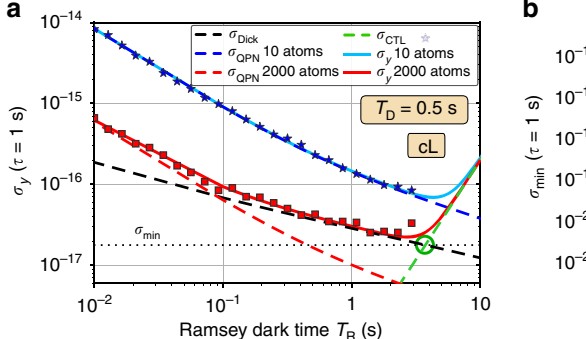
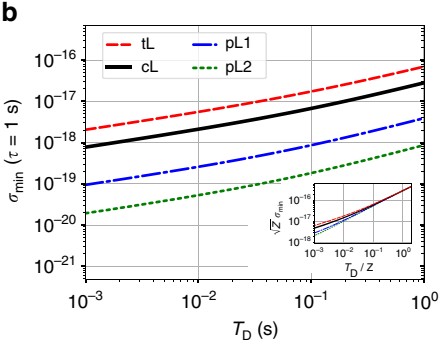

**Fig. 2 Optimal clock stability. a** Allan deviation $\sigma_y$ of an optical atomic clock at averaging time $\tau = 1$ s as a function of Ramsey dark time $T_R$ assuming a dead time $T_D = 0.5$ s and laser noise corresponding to the currently best ultra-stable clock lasers (cL) as characterized in Table 1. Solid lines are instabilities from the full noise model, Eq. (2), with $N = 10$ (blue) and $N = 2000$ (red) uncorrelated clock atoms. Dashed lines show the three contributing noise processes: QPN (blue and red), CTL (green), and the Dick noise (black). Symbols are numerical simulations of the closed feedback loop in agreement with the analytic model until the onset of fringe-hops leads to a sudden, strong increase in instability. **b** Bound on the minimal instability $\sigma_{\text{min}}$ as a function of dead time for four different types of clock lasers (tL, cL, pL1, pL2) as defined in Table 1. Inset: Normalizing by the laser coherence time $Z$ (as defined in the main text and given in Table 1) reveals an almost universal scaling with $\sqrt{Z}\sigma_{\text{min}} = 3.0\times 10^{-16}\,(T_D/Z)^{0.7}$ at longer dead times. We use the transition frequency $\nu_0 \approx 429.228$ THz of $^{87}$Sr for calculations.

In view of Fig. 2a (which concerns uncorrelated atoms in coherent spin states) several observations can be made: First, the instability will attain a minimum for a certain interrogation time. We assume in the following that the clock can operate at this optimal time without running into technical problems such as optical path length fluctuations and others. Second, an important distinction has to be made with regard to the particle number $N$. For small ensembles, where QPN dominates over the Dick effect, the minimal instability is set by a tradeoff between QPN and the CTL (cf. blue line in Fig. 2a). This minimum depends on $N$. However, for large ensembles, where the Dick effect dominates over QPN, the minimal instability is set by a tradeoff between the Dick effect and the CTL (cf. red line in Fig. 2a). This minimum does not depend on $N$ and is determined only by laser noise and dead time. Minor deviations result from details of the feedback loop, gain factor and measurement contrast. In particular there exists a time $T_R^*$ where both of these processes contribute equally, i.e., $\sigma_{\mathrm{Dick}}|_{T_R^*} = \sigma_{\mathrm{CTL}}|_{T_R^*} \equiv \sigma_{\min}$, cf. green circle in Fig. 2a. This sets a lower bound for the combined Allan deviation $\sigma_y \geq \sigma_{\min}$ which is independent of the size $N$ of the ensemble of clock atoms. How closely this bound can be saturated depends on how exactly $\sigma_{\mathrm{Dick}}$ and $\sigma_{\mathrm{CTL}}$ scale with $T_R$. However, in the worst case $\sigma_{\min}$ lies only a factor $\sqrt{2}$ below the true minimum. In Fig. 2b we show the minimal instability $\sigma_{\min}$ as a function of dead time $T_D$ for four types of lasers, as summarized in Table 1: The second is the currently best laboratory clock laser (cL) which is limited by flicker frequency noise at an Allan deviation $\sigma_{\mathrm{FF}} = 4.9 \times 10^{-17}$[59]. We also consider two future generation clock lasers with projected improved noise spectra limited by $\sigma_{\mathrm{FF}} = 10^{-17}$ or $3 \times 10^{-18}$ which we refer to as pL1 and pL2, respectively. Such lasers require vast improvements over state-of-the-art systems. They could possibly be achieved in a combination of low temperature cryogenic systems with pure crystalline components of the cavity, which are capable of achieving a fundamental noise limit in the low $10^{-18}$ range[60]. For comparison, we also include a laser for transportable atomic clocks (tL) whose stability is reduced due to shorter cavities and stronger environmental perturbations compared to the laboratory setting[61]. We consider here an ambitious design, limited by flicker frequency noise at $\sigma_{\mathrm{FF}} = 10^{-16}$, see Table 1 for the detailed characterizations. For all types of lasers an almost universal behavior emerges, as shown in the inset of Fig. 2b, on re-scaling $T_D$ and $\sigma_{\min}$ by the laser coherence time $Z$. Deviations are likely due to the complicated dependence of $\sigma_{\mathrm{Dick}}$ on the duty factor $T_R/T_C$. Note that at $T_D < 10^{-3}$ s contributions to the Dick effect from neglected technical high frequency noise at Fourier frequencies $\geq 1$ kHz can become significant compared to the noise sources considered here. The laser coherence time is defined here by $\sigma_{\mathrm{LO}}(Z) 2\pi\nu_0 Z = 1$ rad following[36], and given in Table 1 for the four types of clock lasers.

**Critical ensemble size.** So far, all statements referred to uncorrelated atoms. Provided we perform Ramsey interrogation of a

single ensemble of atoms, under which conditions can the clock stability be improved by employing squeezed spin states? First, it is clear that the limitation due to dead time in form of the Dick effect will not be reduced by atomic correlations. On the contrary, additional preparation time may even lead to an increase in instability there. Strongly squeezed or other highly entangled states will result in a more restrictive CTL and are unfavorable also for several other reasons (stronger decoherence, unfeasible requirements on measurements etc.). Therefore we consider here only moderately squeezed states which maintain the fringe width and contrast, leaving the CTL largely at the level of coherent states[57]. Specifically, we assume states generated via the unitary one-axis twisting interaction $e^{-i(\mu/2)S_z^2}$ for which the squeezing strength, $\mu \approx 1.1\,N^{-2/3}$, was independently optimized to give the lowest instability for a given particle number $N$ in the dead time free case. The resulting optimal spin squeezing is $\xi^2 = \mathcal{O}(N^{-2/3})$. Further improvements using the one-axis twisting states would need modifications of the protocols with e.g., additional control interactions or nonlinear measurements. We thus arrive at the important conclusion that—with CTL and Dick noise being unchanged—the combined instability is limited by $\sigma_{\min}$, independently of the degree of squeezing. This limit will eventually be met when the QPN is reduced below $\sigma_{\min}$ either by means of spin squeezing (reducing $\xi$) or using a larger ensemble of atoms. Figure 3a shows the Allan deviation versus particle number for various levels of dead time achieved with coherent spin states (CSS) and optimized spin squeezed states (SSS). For sufficiently large ensembles, CSS and SSS approach the same limit given by $\sigma_{\min}$. We infer that, especially for large ensembles, squeezing can provide a gain in stability only for quite challenging levels of dead time. The critical number of particles $N_{\min}$, which is required to achieve the minimal instability for a given dead time, laser stability, and degree of squeezing $\xi$, is set by the condition that the QPN dives below $\sigma_{\min}$, that is $N_{\min} = \min_N\{\sigma_{\mathrm{QPN}}|_{N,T_R^*} \leq \sigma_{\min}\}$. Note that for $T_D = 0$, the Dick effect's contribution in Eq. (2), being the only one that cannot be reduced by larger $N$, vanishes and the definition of $N_{\min}$ is no longer meaningful. In that case one should for any number employ weakly squeezed states as the tradeoff in Eq. (2) is between QPN and CTL only. In Fig. 3b we show $N_{\min}$ for uncorrelated particles (full lines) and squeezed states (dashed) versus $T_D$. At small dead times this shows the expected significant reduction $N_{\min}^{(\mathrm{SSS})} < N_{\min}^{(\mathrm{CSS})}$ for squeezed states, which results from the reduction of the squeezing parameter. We conclude that an increased stability using spin squeezed states is only possible in small ensembles with particle numbers $N < N_{\min}^{(\mathrm{CSS})}$ for a given $T_D$ and laser noise. This result highlights how the envisioned improvements in the laser coherence time would make larger ensembles or squeezed states in lattice clocks necessary eventually. In order to assess the long-term perspectives of squeezed states, we show in Fig. 3c the critical particle number $N_{\min}$ as a function of laser instability. The comparison of the two curves shows a slowly increasing separation with reduced

**Table 1 Laser parameters.**

| Laser type | Abbr. | $\sigma_W$ | $\sigma_{FF}$ | $\sigma_{RW}$ | $Z$ [s] |
|---|---|---|---|---|---|
| Projected transportable clock laser | tL | $5.2 \times 10^{-17}$ | $1.0 \times 10^{-16}$ | $2.6 \times 10^{-18}$ | 3.6 |
| Current record laboratory clock laser | cL | $2.5 \times 10^{-17}$ | $4.9 \times 10^{-17}$ | $1.3 \times 10^{-18}$ | 7.5 |
| Projected laboratory clock laser 1 | pL1 | $5.2 \times 10^{-18}$ | $1.0 \times 10^{-17}$ | $2.6 \times 10^{-19}$ | 36.5 |
| Projected laboratory clock laser 2 | pL2 | $1.6 \times 10^{-18}$ | $3.0 \times 10^{-18}$ | $7.8 \times 10^{-20}$ | 118.8 |

Specification of the four types of clock lasers considered in detail. Stability is given in terms of white ($\sigma_W$), flicker ($\sigma_{FF}$) and random walk ($\sigma_{RW}$) frequency noise at an averaging time $\tau = 1$ s. Then with $\nu_0 = 429.228$ THz the coherence time $Z$ is determined as defined in the main text.

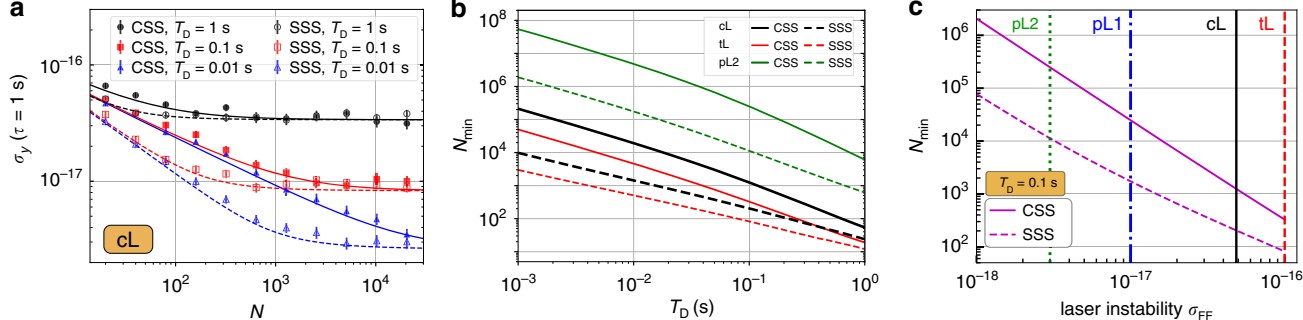

**Fig. 3 Approaching the optimal stability and scaling of the critical particle number. a** Particle number scaling towards the lower bound $\sigma_{min}$ for $T_D = 1$ s, 0.1 s, 0.01 s. The stability for each $N$ is optimized over the Ramsey time. Compared are uncorrelated atoms in a coherent spin state (CSS, full lines and symbols) and squeezed spin states (SSS, dashed lines and empty symbols). Error bars are one standard deviation. **b** Minimal required particle number for reaching the stability limit with uncorrelated particles (full lines) or spin squeezed initial states (dashed lines). **c** Increase of the critical particle number $N_{min}$ with improved laser stability for coherent spin states and squeezed spin states. Vertical lines show the four highlighted laser types described in the main text. The labels (tL, cL, pL1, pL2) denote the respective laser phase noise, as specified in Table 1, in all parts of the figure. We use the transition frequency $\nu_0 \approx 429.228$ THz of $^{87}$Sr for calculations.

instability. This predicts a significant reduction of the required particle numbers when using squeezed states only at high laser quality. Thus, our model also allows to identify concrete conditions of laser stability, from which point on squeezing becomes relevant even for relatively larger ensembles as used in lattice clocks. However, the required laser stability goes far beyond the currently best technical achievements (vertical solid line) and requires considerable improvement of the clock lasers (corresponding to green or blue dashed lines). Finally, the results presented above may be altered if there exists some additional process which places an upper bound $T_{max}$ to the Ramsey time. This could occur due to coherence losses from collisions, photon scattering, a limited natural lifetime or others. Of course, in the case $T_R^* \le T_{max}$, where $T_{max}$ is larger than the optimal interrogation time $T_R^*$ identified above, our results are unchanged. $T_R^*$ is on the order of a few seconds for cL, see supplementary note 1. When $T_R^* \ge T_{max}$ one can define the critical particle number $\tilde{N}_{min}(T_{max}) = \min_N \{ \sigma_{QPN}|_{N,T_{max}} \le \sigma_{Dick}|_{T_{max}} \}$. For example, at $T_D = 0.1$ s and assuming the laser cL, we find that the critical particle number changes only from $N_{min} = 1244$ to $\tilde{N}_{min}(T_{max} = 0.1 \text{ s}) = 3504$, $\tilde{N}_{min}(T_{max} = 1 \text{ s}) = 1475$ and remains unchanged if $T_{max} > 2.6$ s.

The logic presented above neglects the effects of fringe hops which might preclude a stable clock operation at the optimal Ramsey time $T_R^*$ for a clock comprised of $N_{min}$ atoms. To assure the validity of our results we therefore compare $T_R^*$ with the Ramsey time $T_{FH}$ at which fringe-hops appear with probability 1 per total number of clock cycles (~$10^6$ in the numerical simulations performed here). We are able to determine $T_{FH}$ by extending the stochastic differential equation formalism (see "Methods" section) to an equivalent Fokker–Planck equation. From this, a mean first escape time for the phase of the stabilized laser can be calculated. We find that fringe-hops occur when the escape time from the interval $[-\pi, \pi]$ reaches the total number of clock cycles. Our results are in agreement with a previous phenomenological guide $T_{FH} = (0.4 - 0.15N^{-1/3})Z$[36]. In this way we found that the minimal instability can actually be achieved prior to being limited by fringe-hops, i.e., $T_R^* < T_{FH}$, with exceptions only in less relevant regimes of short laser coherence times and long dead times, as shown in the "Methods" section.

**Details on laser noise.** In the last part of this article we will provide more technical background for the description of the

Dick effect and the CTL. The Dick noise, for Ramsey interrogation with infinitely short $\pi/2$ pulses, is[56]

$$\sigma_{Dick}^2(\tau) = \frac{1}{\tau} \frac{T_C^2}{T_R^2} \sum_{k=1}^{\infty} S_{LO,y}(k/T_C) \frac{\sin^2(\pi k T_R/T_C)}{\pi^2 k^2} \quad (3)$$

where $S_{LO,y}(f)$ is the laser's single-sided fractional frequency noise power spectral density. We assume $S_{LO,y}(f) = \sum_{k=-2}^{0} h_k f^k$ with $h_{-2} = 2.4 \times 10^{-37}$ Hz, $h_{-1} = 1.7 \times 10^{-33}$, $h_0 = 1.3 \times 10^{-33}$ Hz$^{-1}$ for a clock laser which is limited on the relevant time-scales by flicker frequency noise at $\sigma_{FF} = 4.9 \times 10^{-17}$ [59]. To represent lasers of varying quality the entire spectral density is scaled. For modeling the CTL we build on[36,57,58], and infer the instability due to measurement noise and ineffective feedback based on a stochastic differential equation (SDE). The SDE describes the evolution of the stabilized laser frequency, driven by noise from the free-running laser but cyclically corrected using information from the measurements, including projection noise. We review the approach in the "Methods" section, along with results regarding the necessary feedback and ways to include fringe-hops. A perturbative solution of the SDE in powers of the laser phase variance then allows us to describe the effects of finite laser coherence in lowest order. The CTL results as a contribution in third order of the laser phase variance. If the free-running laser stability is dominated by power-law noise (i.e., $\sigma_{LO}^2(\tau) \propto \tau^\gamma$ with $\gamma = -1, 0, 1$ corresponding to white frequency, flicker and random walk frequency noise, respectively) the laser phase variance $V_\phi = \chi(\gamma)(T_R/Z)^{2+\gamma}$ scales at specific powers of $T_R/Z$ and with $\chi(\gamma)$ of order unity. As a main result, the SDE gives $\sigma_{eff}^2(\tau) = V_{m+d} T_C/(2\pi\nu_0 T_R \sqrt{\tau})^2$ where $V_{m+d}$ is the variance of measurement outcomes when the dynamics is affected by laser phase diffusion. As this is a combined effect of measurement noise, leading to QPN, and phase diffusion, leading to the CTL, both contributions are inferred from $V_{m+d}$ as we show below. Based on the SDE model $V_{m+d} = V_0 + V_1 + \mathcal{O}(V_\phi^4)$ with[58]

$$V_0 = \frac{\Delta S_y^2}{\langle S_x \rangle^2} + \frac{\Delta S_x^2}{\langle S_x \rangle^2} V_\phi + \frac{3(1-c)^2}{8} \frac{\Delta S_y^2}{\langle S_x \rangle^2} V_\phi^2 \quad (4)$$

and

$$V_1 = (1/6 - c/2 + 4c^2/9) V_\phi^3. \quad (5)$$

Here $c = g\langle S_x \rangle/N$ and $g$ is the gain factor of an integrating servo in the feedback loop, see "Methods" section. This holds for

Ramsey interrogation with weakly squeezed initial states, where measurement statistics are approximated by Gaussian distributions (see "Methods" section). Now, $\sigma_{\text{eff}}^2$ can be separated in the following way: All terms in $V_0$ contain spin variances $V_0 = \xi^2/N$ in the limit $T_R \ll Z$, reproducing the QPN, so $\sigma_{\text{QPN}}^2(\tau) = V_0 T_C/(2\pi\nu_0 T_R \sqrt{\tau})^2$. The CTL is $\sigma_{\text{CTL}}^2(\tau) = V_1 T_C/(2\pi\nu_0 T_R \sqrt{\tau})^2$ as $V_1$ is the first order with an $N$-independent contribution. This term results conceptually from the lowest order (cubic) non-linearity of the sinusoidal Ramsey signal.

## Discussion

In conclusion, we would like to emphasize that the theoretical and experimental progress in manipulating the QPN in quantum metrological measurements with entangled states represents an important and exciting challenge. In the context of atomic clocks, however, a reduction in the QPN does not automatically mean an improvement in statistical uncertainty. A possible gain through entangled states therefore requires an evaluation that is detailed to the specific conditions of an atomic clock. Frequency estimation using GHZ states, which is limited by QPN and atomic decoherence, was already considered quite some time ago in[35]. Here, we have extended this idea to discuss the stability of optical atomic clocks with squeezed states. The model we developed allows a comprehensive and quantitative investigation, in which parameter regimes laser noise is not the most stringent limitation, so squeezing can improve the stability, and in which cases laser noise is dominant and needs to be overcome by other means before squeezing provides an advantage. Although we showed that current improvements are limited to small systems only, our results also indicate that after challenging improvements in laser stability and dead time, spin squeezing will become relevant for optical lattice clocks as well. In order to promote the use of entanglement in optical clocks, a number of further aspects should be considered in a similar way: Excess anti-squeezing due to imperfect state preparation has been considered in ref.[37], and shown to reduce clock stability for white frequency noise. It would be desirable to incorporate excess anti-squeezing to our model which deals with realistic colored laser noise. To what extent other measurement methods besides Ramsey interrogation are subject to similar restrictions or in which cases they can be circumvented remains open. Rabi interrogation is not expected to give improvements over the limits presented here due to its increased QPN and enhanced Dick effect[62], even though it allows for longer interrogation times than Ramsey protocols. The limitations described here, valid for single ensemble clocks with cyclic Ramsey interrogation and dead time, may be overcome with more sophisticated clock architectures: The laser coherence limit can be tackled with adaptive measurement schemes[63] or cascaded systems with multiple ensembles of atoms[41–43]. However, we suspect that including dead time to these studies would still show the existence of a critical ensemble size, limiting the useful regime of squeezed states, similar to what was presented here. Although one should note that the overall stability would improve on what we have presented. Dead time free laser stabilization, basically eliminating the Dick effect, was constructed by anti-synchronized interrogations of two atomic clocks[46]. It is then expected that spin squeezing will again increase the stability for any $N$ but comes at the cost of keeping low systematic shifts for two ensembles. While the underlying method has been demonstrated, showing an improvement through squeezed states remains an open challenge in this setting.

Conceptually different approaches that may evade the presented limitations when applied without dead time are based on continuously tracking the atomic phase via weak measurements[64–66].

## Methods

We introduce a model for optical atomic clocks based on formulating the time evolution of the stabilized differential phase between laser and atomic reference in terms of a stochastic differential equation (SDE) originally proposed in refs.[57,58]. Based on this we discuss the effects of using a two-stage integrating servo to correct out local oscillator fluctuations for all noise types considered here. Afterwards, we review therein how the nonlinear SDE can be solved approximately to generate an expression for the resulting clock instability in orders of the phase variances and finally we discuss the onset of fringe-hops and motivate a possible description via the mean first passage time.

In the following we always consider an optical atomic clock which operates in repeated, identical cycles of duration $T_C$. Each cycle contains a Ramsey dark time $T_R \equiv T$, as well as some dead time $T_D = T_C - T$. Three frequencies are relevant to describe the clock operation: (i) The ideal atomic transition frequency $\nu_0$, which we assume is constant for all times. (ii) The free-running laser frequency $\nu_{\text{LO}}(t)$ for which the stochastic fractional frequency noise is described by a noise power spectral density

$$S_{\text{LO},y}(f) = h_\gamma f^\gamma \tag{6}$$

with $\gamma = -2, -1, 0$ depending on the nature of temporal correlations we wish to study. (iii) The stabilized laser frequency $\nu(t)$ which results from the periodic feedback corrections on the free running laser frequency based on the error signal derived from probing the atomic ensemble.

In order to derive an effective measurement variance, which describes the long term stability of the clock, we first discuss the evolution of the stabilized frequency between the Ramsey times of each cycle. The average stabilized frequency difference during the Ramsey dark time of cycle $k$ is

$$\delta\nu_k = \frac{1}{T} \int_{(k-1)T_C}^{(k-1)T_C+T} [\nu(t) - \nu_0] \mathrm{d}t \tag{7}$$

and gives rise to a differential phase

$$\phi_k = 2\pi \int_{(k-1)T_C}^{(k-1)T_C+T} [\nu(t) - \nu_0] \mathrm{d}t = 2\pi\,\delta\nu_k\,T \tag{8}$$

before the measurement at time $(k-1)T_C + T$. Due to the recursive nature of the feedback correction, the stabilized frequency difference can be split into

$$\delta\nu_k = \delta\nu_k^{\text{LO}} - p_{k-1}. \tag{9}$$

The first term, $\delta\nu_k^{\text{LO}}$, is the average frequency difference contributed by the free-running laser, whereas $p_{k-1}$ is the frequency correction of the servo applied at the end of the previous cycle. For the differential phase,

$$\phi_k = 2\pi\delta\nu_k T = 2\pi\delta\nu_k^{\text{LO}} T - 2\pi p_{k-1} T \tag{10}$$

applies accordingly. The specific form of the correction $p_{k-1}$ depends on the choice of the servo. A frequently used method of feedback is to have an integrator as the servo. In this case, the correction to the laser frequency is constructed as

$$p_{k-1} = \frac{g}{2\pi T}\left(\hat{\phi}_{k-1} + \frac{2\pi T}{g}p_{k-2}\right) = \frac{g}{2\pi T}\hat{\phi}_{k-1} + p_{k-2}. \tag{11}$$

Here $g$ is the gain factor and $\hat{\phi}_{k-1}$ an estimator for the accumulated phase during the Ramsey interrogation based on the measurement result (in the simplest case the estimator is just the measurement result itself). From this one finds the coupled stochastic difference equations

$$\delta\nu_k - \delta\nu_{k-1} = \delta\nu_k^{\text{LO}} - \delta\nu_{k-1}^{\text{LO}} - \frac{g}{2\pi T}\hat{\phi}_{k-1} \tag{12}$$

$$\phi_k - \phi_{k-1} = \phi_k^{\text{LO}} - \phi_{k-1}^{\text{LO}} - g\hat{\phi}_{k-1} \tag{13}$$

for average frequency and phase. We now focus on the phase Eq. (13). The estimate therein stems from a measurement outcome of the collective spin $\cos(\phi_{k-1})S_y + \sin(\phi_{k-1})S_x$. This holds for standard Ramsey interrogation which applies the sequence $e^{-i\pi/2S_x}e^{-i\phi_{k-1}S_z}$ of interactions onto an initial state before measuring $S_z$. Here we consider either an uncorrelated state or a spin squeezed state generated via one-axis twisting, as the input state. Both states are polarized dominantly in $x$-direction. Working in the small squeezing strength regime, we are able to approximate the stochastic measurement outcomes by Gaussian random variables. As $\langle S_x S_y + S_y S_x \rangle = 0$ for both types of states, we can further separate the measurement outcomes as a linear combination of two independent Gaussian random variables describing the measurement results of $S_x$ and $S_y$ correspondingly. The statistics of these squeezed states are well known. When the reduced variance is aligned with the measurement direction $y$ one finds[15]

$$\frac{\Delta S_y^2}{\langle S_x\rangle^2} = \frac{1}{N}\frac{1 + \frac{1}{4}(N-1)(A - \sqrt{A^2 + B^2})}{\cos^{2N-2}(\mu/2)} \tag{14}$$

and

$$\frac{\Delta S_x^2}{\langle S_x\rangle^2} = \frac{1}{N}\frac{N(1 - \cos^{2(N-1)}(\mu/2)) - (N/2 - 1/2)A}{\cos^{2N-2}(\mu/2)} \tag{15}$$

with $A = 1 - \cos^{N-2}(\mu)$, $B = 4\sin(\mu/2)\cos^{N-2}(\mu/2)$. The measurement contrast decays as

$$\langle S_x\rangle = \frac{N}{2}\cos^{N-1}(\mu/2). \tag{16}$$

When the overall measurement outcome of the Ramsey interrogation is identified as the phase estimate from above, one finds

$$\hat{\phi}_{k-1} = \frac{\kappa}{T}\sin(\phi_{k-1})dt + \frac{\kappa}{\sqrt{T}}\frac{\Delta S_x}{\langle S_x\rangle}\sin(\phi_{k-1})\,dW_{x,k} + \frac{\kappa}{\sqrt{T}}\frac{\Delta S_y}{\langle S_x\rangle}\cos(\phi_{k-1})\,dW_{y,k} \tag{17}$$

where $\kappa = \langle S_x\rangle/S$ quantifies the measurement contrast and at this stage $dW_{(x,\,y),k}$ are random numbers with a standard normal distribution, representing fluctuation of the $k$th measurement outcome for $S_{x,y}$ around their mean values. The differential time increment $dt = T$ corresponds to the Ramsey duration as we are interested in studying the phase evolution over the course of many interrogation cycles. When going to time continuous stochastic differential equations, $dW_{x,y}$ will then be standard Wiener processes adding measurement noise, hence the notation. We also did not cancel the factors $T$ in the first term on the right-hand side to highlight the correspondence to the continuous stochastic differential equation. In Eq. (17) we used the measured phase as the linear estimate of the actual phase value which is a good choice for small phases in each Ramsey interrogation but could in principle be improved through nonlinear estimation from the measurement result. The above form, where formally $T$ could be removed from the first summand, is motivated by our overall goal to develop a stochastic differential equation for the time evolution of the stabilized phase.

As a first result we now show that the single integrator described above is not sufficient to suppress all laser noise types we consider even under otherwise ideal conditions. Especially for stronger temporal correlations, as is the case for random-walk of frequency, this choice of the servo can not fully correct out all fluctuations. This has been a shortcoming in a previous, mathematically more rigorous, approach[67] leading to lower bounds on the stability that are expected to hold for white frequency and flicker frequency noise but are not fundamental for random walk or more strongly correlated noise of the local oscillator. In that case, the limits can be overcome with a different choice of servo. Modifying the servo is easily possible in the difference equations by adapting the servo correction. Consider now a double-integrator with

$$p_{k-1} = p_{k-2} + \frac{g}{2\pi T}\hat{\phi}_{k-1} + \frac{g_2}{2\pi T}\sum_{n=1}^{k}\hat{\phi}_{k-n} \tag{18}$$

including longer averages of estimates with the secondary gain factor $g_2 \ll g$. Such secondary integrator stages already find applications in the operation of atomic clocks to also counteract slow drifts of the laser frequency[68]. Alternatively, servos employing optimized general linear predictors have also been considered in the literature[36,69]. The effect on the stochastic difference equation is straightforward. To more clearly see the suppression of noise via the double integrator, we transform the finite stochastic difference Eq. (13) into a system of stochastic differential equations (note that this disregards the dead times now):

$$d\phi = d\phi_{LO} - g\frac{\kappa}{T}\phi\,dt - g\frac{\kappa}{T^{1/2}}\frac{\Delta S_y}{\langle S_x\rangle}\,dW_y(t) - \frac{g_2}{T}\psi\,dt \tag{19}$$

$$d\psi = \frac{\kappa}{T}\phi\,dt + \frac{\kappa}{T^{1/2}}\frac{\Delta S_y}{\langle S_x\rangle}\,dW_y(t) \tag{20}$$

where functions of $\phi$ in Eq. (17) were expanded only up to linear order and non-linear terms involving products different variables are neglected for now.

At this point, we would like to emphasize that neglecting higher orders in $\phi$ can only be justified for small phase variations. However, if the instability of the atomic clock is to be optimized over the Ramsey time, these terms must be considered. The increase of in instability with $T$, the coherence time limit (CTL), is substantial for the results of the main text. Therefore, all numerical results we refer to come from simulations of the stochastic difference equations

$$\delta\nu_k - \delta\nu_{k-1} = \delta\nu_k^{LO} - \delta\nu_{k-1}^{LO} - \frac{g}{2\pi T}\hat{\phi}_{k-1} \tag{21}$$

$$\phi_k - \phi_{k-1} = \phi_k^{LO} - \phi_{k-1}^{LO} - g\hat{\phi}_{k-1} \tag{22}$$

with the true (non-Markovian) local oscillator noise and with the full (non-linear) phase estimation as given in Eq. (17). The only difference between Eqs. (21), (22) and (12), (13) are the terms proportional to $g_2$ from applying the double integrator as stated in Eq. (18). How the analytic expression for the CTL follows from the stochastic differential equation without linear approximation is discussed further below.

In addition, the variable $\psi(t)$ was introduced which in general is

$$\psi(t) = \int_0^t \left[\frac{\kappa}{T}\sin(\phi(t'))dt' + \frac{\kappa}{\sqrt{T}}\frac{\Delta S_x}{\langle S_x\rangle}\sin(\phi(t'))\,dW_x(t') + \frac{\kappa}{\sqrt{T}}\frac{\Delta S_y}{\langle S_x\rangle}\cos(\phi(t'))\,dW_y(t')\right]. \tag{23}$$

The system of differential Eqs. (19) and (20) can be expressed as

$$d\boldsymbol{w} = M\boldsymbol{w}\,dt + d\boldsymbol{f}(t) \tag{24}$$

with

$$\boldsymbol{w}(t) = \begin{pmatrix}\phi(t)\\\psi(t)\end{pmatrix}, \quad M = \begin{pmatrix}-g\frac{\kappa}{T} & \frac{-g_2}{T}\\\frac{\kappa}{T} & 0\end{pmatrix}, \quad d\boldsymbol{f}(t) = \begin{pmatrix}d\phi_{LO}(t) - g\frac{\kappa}{T^{1/2}}\frac{\Delta S_y}{\langle S_x\rangle}dW_y(t)\\\frac{\kappa}{T^{1/2}}\frac{\Delta S_y}{\langle S_x\rangle}dW_y(t)\end{pmatrix}. \tag{25}$$

Equation (24) can be solved formally via Fourier transform resulting in

$$\boldsymbol{w}(\omega) = (i\omega\mathbb{1} - M)^{-1}\boldsymbol{g} \tag{26}$$

where

$$\boldsymbol{g} = \begin{pmatrix}i\omega\phi_{LO}(\omega) - g\frac{\kappa}{T^{3/2}}\frac{\Delta S_y}{\langle S_x\rangle}\,dW_y(\omega)\\\frac{\kappa}{T^{3/2}}\frac{\Delta S_y}{\langle S_x\rangle}\,dW_y(\omega)\end{pmatrix}. \tag{27}$$

Based on the solution (Eq. 26) we calculate the spectrum

$$S_w(\omega) = \langle\boldsymbol{w}(\omega)\boldsymbol{w}^\dagger(\omega)\rangle = (i\omega\mathbb{1} - M)^{-1}\boldsymbol{g}\boldsymbol{g}^\dagger(-i\omega\mathbb{1} - M^\dagger)^{-1} \tag{28}$$

and find as a part of this the spectrum of the stabilized phase

$$S_\phi(\omega) = \frac{S_{LO}(\omega) + \left[\frac{g_2^2\kappa^2}{\omega^4 T^5} + \frac{g^2\kappa^2}{\omega^2 T^3}\right]\frac{\Delta S_y^2}{\langle S_x\rangle^2}S_{dW_y}(\omega)}{1 + (g^2\kappa^2 - 2g_2\kappa)/(\omega T)^2 + g_2^2\kappa^2/(\omega T)^4} \tag{29}$$

where we used that the laser noise $\phi_{LO}$ is independent from the atomic measurement results $dW_y$. As we are interested in the final long term stability of atomic clocks, at $\tau \gg T_C$, we thus expand Eq. (29) in lowest orders of $\omega$. In the limit $\omega T/g_2 \ll 1$, reached at low enough Fourier frequencies $\omega$ for any given values of $T$ and $g_2$, this reduces at first to

$$S_\phi(\omega) = \frac{(\omega T)^4}{g_2^2}S_{LO}(\omega) + \frac{1}{T}\left[1 + \frac{g^2}{g_2^2}(\omega T)^2\right]\frac{\Delta S_y^2}{\langle S_x\rangle^2}. \tag{30}$$

Equation (30) shows that local oscillator noise is suppressed for all noise correlations considered in this work, i.e., a scaling of the spectral density with $S_{LO}(\omega) \propto 1$, $S_{LO}(\omega) \propto 1/\omega$ and even $S_{LO}(\omega) \propto 1/\omega^2$. Furthermore for all these local oscillator correlations the dominant contribution at long averaging times ($\omega \to 0$) will be the white atomic noise

$$S_\phi(\omega) = \frac{\Delta S_y^2}{T\,\langle S_x\rangle^2}. \tag{31}$$

However, we note again that this only holds under linear approximation in the stochastic differential equation.

From these results we argue that the double integrating servo completely corrects frequency errors and removes correlations between phases in different measurement cycles. Therefore, we approximate from now on the local oscillator driven phases $d\phi_{LO}$ as uncorrelated Wiener increments with a scaling of the variance $V_\phi = \chi(\gamma)(T_R/Z)^{2+\gamma}$ that is appropriate to the specific noise type within an individual Ramsey dark time, with $\chi = 1, 1.7, 2$ for $\gamma = -1, 0, 1$. Overall, we approximate the stochastic difference Eq. (13) (with Eq. (17)) for the phase estimate $\hat{\phi}_{k-1}$ by the single stochastic differential equation

$$d\phi = \sqrt{V_\phi}d\phi_{LO} - g\frac{\kappa}{T}\sin(\phi)dt - g\frac{\kappa}{\sqrt{T}}\frac{\Delta S_x}{\langle S_x\rangle}\sin(\phi)dW_x - g\frac{\kappa}{\sqrt{T}}\cos(\phi)\frac{\Delta S_y}{\langle S_x\rangle}dW_y. \tag{32}$$

An approximate solution to this non-linear SDE can be constructed from a power series ansatz[58]

$$\phi(t) = \sum_{n=1}^{\infty}\epsilon_1^n\phi_{1,n}(t) + \epsilon_2\sum_{n=0}^{\infty}\epsilon_1^n\phi_{2,n}(t) + \epsilon_3\sum_{n=0}^{\infty}\epsilon_1^n\phi_{3,n}(t) \tag{33}$$

assuming small perturbation parameters $\epsilon_1 = \sqrt{V_\phi}$, $\epsilon_2 = \frac{\Delta S_y}{\langle S_x\rangle}$ and $\epsilon_3 = \frac{\Delta S_x}{\langle S_x\rangle}$. Here the variance $V_\phi$ quantifies again the width of the phase distribution prior to each

measurement. The greater the correlations in the laser noise, the faster the width $V_\phi$ of the phase distribution increases with the Ramsey time. For details on the further steps and the calculation of the Allan variance in the case of linear feedback we refer to ref. [58] but note that we also included here a term proportional to $\epsilon_2\epsilon_1^2$ not treated in the reference. The result to lowest orders in the perturbation parameters is

$$\sigma_{\text{eff}}^2(\tau) = \frac{1}{(2\pi\nu_0 T_R)^2}\frac{T_C}{\tau}V_{\text{m+d}} \tag{34}$$

with

$$V_{\text{m+d}} = \frac{\Delta S_y^2}{\langle S_x\rangle^2} + \frac{\Delta S_x^2}{\langle S_x\rangle^2}V_\phi + \frac{3(1-c)^2}{8}\frac{\Delta S_y^2}{\langle S_x\rangle^2}V_\phi^2 \\ + \frac{1-3c+8c^2/3}{6}V_\phi^3 + \mathcal{O}(V_\phi^4) \tag{35}$$

and

$$c = g\frac{\langle S_x\rangle}{N} \tag{36}$$

as applied in the main text.

Finally, it is worth noting that this model, evaluating the Allan variance, often does not correctly reflect the appearance of fringe-hops. Upper limits for safe Ramsey times, within which less than 1 fringe-hop per $10^6$ clock cycles occurs, have so far only been determined by numerical simulations of the full stochastic process[36]. According to that study,

$$T_{\text{FH}} = (0.4 - 0.15N^{-1/3})Z \tag{37}$$

and

$$T_{\text{FH}} = (0.4 - 0.25N^{-1/3})Z \tag{38}$$

were suggested as guides for safe interrogation times in the case of flicker frequency and random walk of frequency noise, respectively. As described in the main text, this guide can be used to estimate for which parameters fringe-hops may occur before reaching the intersection of Dick effect and CTL. Figure 4 shows corresponding parameter landscapes illustrating the relation of the two time scales, $T_{\text{FH}}$ and $T_R^*$, against laser coherence time $Z$ and dead time $T_D$. It can also be seen that the region with $T_{\text{FH}} < T_R^*$ decreases for increasing particle numbers.

In contrast to the numerically motivated guides above, the onset of fringe-hops can also be predicted by further investigations of the SDE as outlined below. This may also allow a better understanding of the underlying processes in the future. First, we observe that the SDE (Eq. 32) may be expressed more compactly as

$$d\phi = A(\phi)dt + \boldsymbol{b}(\phi)\cdot d\boldsymbol{W}(t) \tag{39}$$

with

$$A(\phi) = -g\frac{\kappa}{T}\sin(\phi), \tag{40}$$

and the coefficients

$$\boldsymbol{b}(\phi) = \left(\sqrt{V_\phi}, -g\frac{\kappa}{\sqrt{T}}\frac{\Delta S_x}{\langle S_x\rangle}\sin(\phi), -g\frac{\kappa}{\sqrt{T}}\cos(\phi)\frac{\Delta S_y}{\langle S_x\rangle}\right)^T \tag{41}$$

to the 3 independent Wiener processes

$$d\boldsymbol{W} = (d\phi_{\text{LO}}, dW_x, dW_y)^T. \tag{42}$$

Generally, a stochastic differential equation of this form can be rewritten into an equivalent Fokker–Planck equation[70], which in this case reads

$$\partial_t P(\phi,t) = \left[-\partial_\phi A(\phi) + \frac{1}{2}\partial_\phi^2 B(\phi)\right]P(\phi,t) \tag{43}$$

with

$$A(\phi) = -q\sin\phi, \quad B(\phi) = \boldsymbol{b}\cdot\boldsymbol{b} = r + s\cos^2\phi \tag{44}$$

where $q = -g\frac{\kappa}{T}$, $r = V_\phi + g^2\frac{\kappa^2}{T}\frac{\Delta S_x^2}{\langle S_x\rangle^2}$ and $s = g^2\frac{\kappa^2}{T}\left(\frac{\Delta S_y^2}{\langle S_x\rangle^2} - \frac{\Delta S_x^2}{\langle S_x\rangle^2}\right)$. The idea for connecting this to fringe-hops is to consider the so-called mean first passage time (mfpt). The mean first passage time describes the average duration over which a random variable (here the stabilized phase) remains within a given interval. Note that the passage time in this cases is again to be regarded as a multiple of the feedback cycle duration. In order to calculate the mfpt we use established tools of stochastic methods[70]. A useful function in the context of mfpt is

$$\Psi(x) = \exp\left\{\int_0^x dx'\frac{2A(x')}{B(x')}\right\} \tag{45}$$

$$= \exp\left\{\frac{2q}{\sqrt{r\,s}}\left[\arctan\left(\sqrt{\frac{s}{r}}\cos x\right) - \arctan\left(\sqrt{\frac{s}{r}}\right)\right]\right\}. \tag{46}$$

From this, the mean first time to escape the interval $[-a, a]$, assuming the laser phase starts at $\phi = 0$, is given by[70]

$$T_{\text{mfpt}} = \frac{2\left[\left(\int_{-a}^0\frac{dz}{\Psi(z)}\right)\int_0^a\frac{dx}{\Psi(x)}\int_{-a}^x dy\frac{\Psi(y)}{B(y)} - \left(\int_0^a\frac{dz}{\Psi(z)}\right)\int_{-a}^0\frac{dx}{\Psi(x)}\int_{-a}^x dy\frac{\Psi(y)}{B(y)}\right]}{\int_{-a}^a\frac{dz}{\Psi(z)}}. \tag{47}$$

Which can be further simplified to the double integral

$$T_{\text{mfpt}} = \int_{-a}^a dx\int_{-a}^x dy[2\Theta(x) - 1]\frac{\Psi(y)}{\Psi(x)}\frac{1}{B(y)} \tag{48}$$

where $\Theta(x)$ is the Heaviside function, by using the symmetry $\Psi(x) = \Psi(-x)$.

A maximum Ramsey time $T_{\text{FH}}$ without fringe-hops can be specified by requesting that the stabilized phase should not leave the interval $(-\pi, \pi)$, where it is corrected back to the original reference point $\phi = 0$, within the simulated ~$10^6$ cycles of clock operation. So $T_{\text{mfpt}}(T_R) \le 10^6$ for $T_R \le T_{\text{FH}}$, where the functional dependence of the mfpt on $T_R$ is based on the parameters $q$, $r$, $s$ depending on this quantity. In the case of flicker frequency noise this led to a good agreement with the onset of fringe-hops observed in numerical simulations. For random walk noise we achieved slightly better agreements when assuming the interval $(-\pi/2, \pi/2)$ for the calculation of the mean first passage time. We found that this stronger requirement is more applicable here due to the increased temporal correlations which already cause fringe-hops in a regime where the feedback, though insufficient, is not on paper stabilizing to a different fringe. Figure 5 compares $T_{\text{FH}}$ as based on the mfpt with results from numerical simulations of the full clock operation, as well as the phenomenological guides (Eq. 37) and (Eq. 38) in the case of uncorrelated atoms. For both noise types the prediction of the mfpt also exhibits a constant cutoff for large $N$ and reduced $T_{\text{FH}}$ for smaller ensembles which is in qualitative and quantitative agreement with the guides and the numerical results. Except the escape interval, as mentioned above, all calculations are without free parameters. For small ensembles, e.g., $N = 1$, our theory falls short in accurately predicting $T_{\text{FH}}$ as it uses

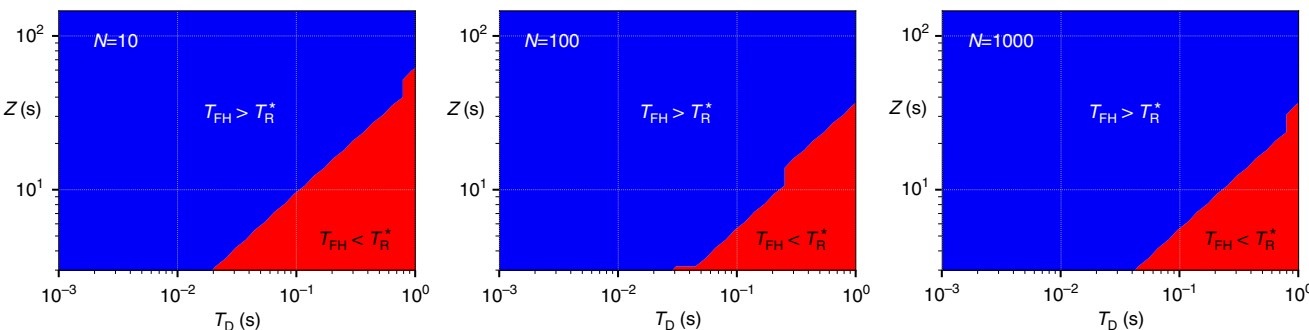

**Fig. 4 Fringe-hops and interrogation time.** Comparison between the safe interrogation times $T_{\text{FH}}$ (without fringe-hops) and the point of intersection for minimal instability as a function of laser coherence time $Z$ and dead time $T_D$. Based on this study we can identify regions in which the minimum between Dick effect and CTL can be safely reached (blue) and regions in which additionally the occurrence of fringe-hops has to be investigated in detail (red). By comparing $N = 10$, 100, and 1000 one finds that the red region is largest for small ensembles, short coherence times and larger dead times and decreases in size with increased particle numbers.

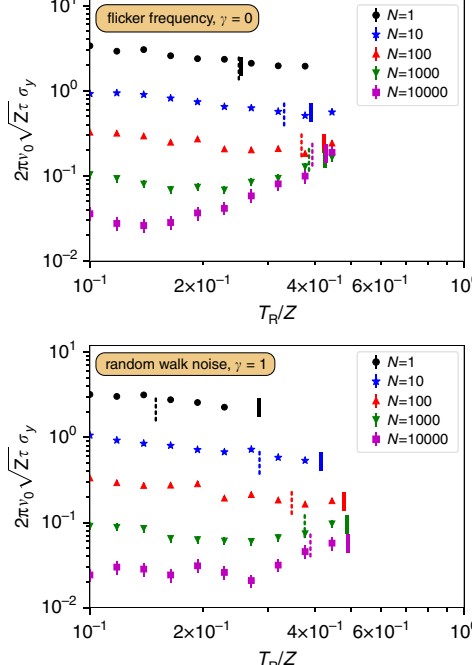

**Fig. 5 Modeling fringe-hops.** Numerically simulated dimensionless clock instability $\sigma_y$ (symbols) as a function of the Ramsey time $T_R/Z$ for a comparison to the predicted onset of fringe-hops, based on a mean first escape time (solid bars), as well as the safe interrogation times (dashed bars) suggested by Eq. (37) and (38). For both, flicker frequency noise and random walk of frequency noise, we find that the predictions based on the mean first escape time reproduce the observed sudden increase in instability well. To include different noise strengths, we normalized all time scales to the laser coherence time $Z$ (see main text). Error bars are one standard deviation from a finite size averaging in the numerical simulations.

the assumption of phases with variance $V_\phi$ for each interrogation, which in this regime is assumed to break down at larger $T_R$.

## Data availability

All data supporting the findings of this study are available from the corresponding authors upon request.

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

## Acknowledgements

We acknowledge valuable contributions from I. D. Leroux in initiating the numerical simulation of atomic clocks applied here. This work is funded by the Deutsche Forschungsgemeinschaft (DFG, German Research Foundation) through CRC 1227 DQ-mat projects A05, A06, B02, B03 and Germany's excellence strategy-EXC-2123 QuantumFrontiers-390837967. P.O.S. and U.S. acknowledge funding from EMPIR under project USOQS. EMPIR projects are co-funded by the European Union's Horizon 2020 research and innovation program and the EMPIR participating states.

## Author contributions

M.S. and K.H. developed the analytic model. M.S. performed the numerical simulations. M.S., C.L., U.S., P.O.S., and K.H. contributed to the development and interpretation of the main results. M.S. and K.H. prepared the manuscript, with input from C.L., U.S., and P.O.S.

## Funding

## Competing interests

The authors declare no competing interests.
