## [Peer Review File · Nature Communications]

Reviewers' Comments:

Reviewer #1:

Remarks to the Author:

Attached

Reviewer #2:

Remarks to the Author:

The authors propose a theoretical study of the use of spin-squeezing for optical clocks. The main point of the paper is that already for a moderate number of atoms, the instability is not limited by the QPN (for an optimal timing of the clock sequence), but rather by the technical noise from the ultra-stable laser sources, even taking into account a significant improvement of these sources.

It is indeed no mystery that taking advantage of spin-squeezing would require a significant work on the laser systems, and I congratulate the authors for quantitatively working this out. The paper is very well written and contain enough details to be usefully reproduced. A significant aspect of the paper is the inclusion in the noise model of the phase diffusion process during the probing of the atoms, which is critical to the limits the authors established.

My main concern comes from the universality of the results presented in the paper, that, in my opinion the authors emphasize too much on, starting from the abstract of the paper. There are several interrogations protocols that have been specifically designed to mitigate the Dick effect and the phase diffusion process (Dead time free clocks, weak measurement, possibly with a combination of several atomic ensembles...) and that the authors barely mention in the sentence:

"The limitation could be avoided with schemes achieving dead-time-free interrogation or overcoming laser phase noise [48–54]. The potential gain from entanglement should then be assessed by an appropriate analysis, incorporating the tradeoffs discussed here."

With a bit of exaggeration, I have the feeling that the authors tautologically claim that if one does not take measures to deal with the clock laser noise, then the clock laser noise is limiting.

Another important aspect that I think is missing (it is only dealt with in the supplementary materials on a specific case) is the duration of the Ramsey interrogation that corresponds to the minimal Allan deviation. There are several practical limitation to this duration that may be difficult to overcome (coherence losses by collisions, photon scattering, natural linewidth...), that may very well alter the maximal number of atoms for which SSS are useful. All in all, I think it would be hard to operate an optical lattice clock with a Ramsey dark time large than typically 1 min. I think it would be of interest that this aspect is incorporated in the graphs of figure 2.

So in conclusion, I think that the paper is extremely useful for the community because of the high quality of the model the authors developed and explained. It is fine if the authors apply this model in a particular interrogation sequence (that is currently typically employed in optical clocks), but I think the main message that the paper conveys (basically that the usefulness of spin squeezing is essentially limited to optical clocks with a rather small number of atom) overcomes what is actually proven in the paper. I think the authors should more quantitatively consider this before publication, or at least make it more clear. The possible restricted universality of the results presented by the authors could limit the interest of the paper to a broad scope journal like nature comm.

Reviewer #3:

Remarks to the Author:

The authors analyze the stability of optical atomic clocks using spin squeezed states in the presence of realistic laser phase noise and dead time in the clock interrogation cycle. Analytic calculations for the asymptotic stability are derived and confirmed with numerical simulations. The results are presented in various parameter space regions including low atom-number and high atom-number as well as better or worse laser phase noise. Based on their analysis, the authors conclude that with current clock laser technology, squeezing only helps clocks with small (order 10) atom number, but for clocks using improved future clock lasers squeezing can help even for large atom numbers.

This analysis will certainly be of interest to the atomic clock and quantum metrology communities, but it's not obvious to me that it is of general enough interest for the audience of Nature Communications. The introduction requires quite a bit of physics background knowledge to be fully understood. The qualitative description filling the first 4 pages of the manuscript is largely already known and various regimes have been described elsewhere, although this is perhaps the most clear and complete description I am aware of. The last page of main text and the methods section contain new analytic calculations, which are quite interesting but may be difficult for a broad audience to follow. Perhaps Nature Physics or PRL would be a better fit?

Specific comments follow, in no particular order:

1. Page 5 - This abbreviated description of the SDE analysis is difficult for me to follow, and I consider myself an expert in this field. I think that people in fields less quantitative than physics will have almost no hope of understanding. It might be better if the discussion of the SDE analysis is entirely moved to the methods (although then the main text would not contain any description of the novel part of this work), or if the entire methods section is incorporated at the end of the main text in a longer article format journal. Some symbols are undefined or use un-descriptive symbols, further compounding the difficulty of understanding (i.e., What does σ_{eff} stand for in σ_{eff} ? Effective doesn't really make sense in this context. Furthermore, it's ambiguous whether σ_{eff} is the total clock instability or just the contribution from CTL, and it's not explicitly defined).
2. Figure 2 (especially the 10 atom case) suggests to me that CTL and fringe hops contribute at a similar level. I say this because the data points for the simulated clock stop (due to fringe hops according to the text) somewhat before T_R^* and the best stability is $\sim 20\%$ above the curve adding QPN, CTL, and Dick effect. This is supported by Figure 4 in the methods which shows that $T_{\text{FH}} < T_R^*$ for 10 ions and 0.5 s dead time. However, the main text (right column of page 2) says that CTL is a "more stringent limit" than fringe hopping. Would it be more accurate to say that CTL and fringe hopping contribute at a similar level?
3. Figures 2 & 3 - The transition frequency of the stimulated clock should be specified, since this is necessary to reproduce the plots.
4. Table 1 - Similarly, the laser frequency should be specified, since the coherence time depends on it.
5. Above equation 13 - It seems rather disjointed to refer the reader to the supplemental information for the description of the statistics, especially since the discussion in the supplement is only one paragraph long. I'd just move section I of the supplement to the location in the methods where it is referenced.
6. Equation 13 - "dt" is not defined.
7. Below equation 27 - Although V_{ϕ} is defined at the end of the main text, it would be useful to remind the reader what it means here.
8. Figure 4 - Where do the kinks in the boundary between the red and blue regions come from? I would have expected this boundary to be smooth.
9. Page 12 - Why is a different interval used for flicker vs random walk? In both cases, the phase estimate is wrong if the actual phase is outside $(-\pi/2, \pi/2)$.
10. Figure 5 - The x and y scales use very small font; it is difficult to read even on a large monitor.

we thank all reviewers for their careful reading of our manuscript and their generally positive evaluation.

Below, we provide a point-by-point response to the referee reports, reply to the expert technical questions posed by the referees, and try to address their concerns.

The feedback by the referees has helped us to improve the manuscript and to better convey its main message.

For all authors,

Marius Schulte

Reviewer #1:

In the article, the authors study the stability of atomic clocks. The paper has three important contributions:

1. Modelling of Atomic Clocks. There is no established mathematical model of atomic clocks. The authors propose and analyze a new model based on the previous works [56, 57]. Having a good concrete model is needed to answer questions about how to improve atomic clocks in the future. For example, the question if squeezing helps can't be answered without detailed calculations. There is a lot of literature about how entangled states can improve the stability of clocks – however, this is far from clear, and as the author show (next point), this crucially depends on the architecture of the clock.
2. Quantitative Predictions of Allan Variance. The authors produce detailed theoretical predictions of Allan variance. As far as I know, this is the first work that gives such theoretical prediction. Based on these predictions, they give concrete answers for the question in which parameter regimes does squeezing help, depending on the dead time, Ramsey time, coherence time of the laser, and the number of atoms. They show that only atomic clocks operating with a small number of atoms can be improved by squeezing. The authors also provide graphs of how much laser stability would need to improve for a given number of atoms for squeezing to be helpful.
3. Fringe Hopping Analysis. The authors give the first theoretical prediction for safe operation regimes without fringe hops that match the previous phenomenological formulas. In particular, they show that the resulting restriction is less severe than the restriction due to CTL noise and hence not important for current atomic clocks.

The second contribution will be used in design of future atomic clocks. The first and third contributions advance our theoretical knowledge of atomic clocks. Overall, I enthusiastically recommend the paper for publication in Nature Communications.

Some comments and questions that might improve the article are below.

Dear Authors,

I really like the work. I think the line of work was missing in the atomic clocks literature. Below are questions and comments in the order of their appearance in the text.

1. (page 1) "To dismiss this noise as a technical imperfection would render the problem trivial." Indeed! It is impossible to answer the questions if (and how much) entanglement helps without incorporating laser noise in the model.

We are glad the referee is also highlighting this essential point, especially in view of the conflicting criticism on exactly this point in the report by Referee #2

2. (page 2) It might not be evident from the wording of the text that [14] is a citation for Eq.(1) and not just for the spin squeezing parameter.

(page 2) We added Reference [14] explicitly before Eq.(1) to clarify this point.

3. What would be the conclusions about the relevance of Fringe hops in the limit of zero dead time? It looks like that the optimal TR is getting smaller if TD is smaller, which should make Fringe hops even less relevant – this seems to be the conclusion of Fig 4. However, if TD is smaller then, I can, I think, squeeze more, which makes the invertible domain smaller. So to restate the question, how much squeezing is helpful if $TD = 0$?

The reviewer is correct in his assessment that low dead time reduces the relevance of fringe hops. We added a statement to section 2 of the supplementary information to expand on this point. However, fringe-hops will remain relevant for very small ensembles, $N < 10$, even at $T_D = 0$ because of the extremely strong influence of the quantum projection noise in that regime. In these cases, the typically non-Gaussian measurement noise also makes accurately modelling of fringe-hops very difficult.

Regarding the second question: In the case when $T_D=0$ it is known from studies of the Wineland squeezing parameter that for conventional Ramsey interrogation only relatively weak squeezing is optimal (see e.g. Pezzè et al, Rev. Mod. Phys. 90, 035005 (2018), Fig. 14). This is what we also find by independently optimizing the squeezing strength. To emphasise this, we revised the following statements on page 4: *"The resulting optimal spin squeezing is $\chi^2 = O(N^{-2/3})$. Further improvements using the one-axis twisting states would need modifications of the interrogation protocols with e.g. additional control interactions or nonlinear measurements."*

To our knowledge, finding optimal squeezing protocols for atomic clocks which extend the conventional Ramsey protocol is still an open problem. Especially for small systems ($N \sim 10$) where fringe-hops remain a big concern.

4. Related question, what is the critical particle number N for $TD = 0$? Is there one?

In short, there is no useful notion of the critical particle number as we define it in the dead time free case. To clarify we added the statement (page 4): “*Note that for $T_D=0$, the Dick effect’s contribution in Eq. 2, being the only one that cannot be reduced by larger N , vanishes and the definition of N_{\min} is no longer meaningful. In that case one should for any number employ weakly squeezed states as the trade-off in Eq.(2) is between QPN and CTL only.*”

5. (page 3) What is the almost universal scaling mentioned in Fig.2 b)?

(page 3) We added “[...] reveals an almost universal scaling with $\sqrt{Z} \sigma_{\min} = 3 \cdot 10^{-16} (T_D/Z)^{0.7}$ s at longer dead times.” in the caption for Figure 2.

6. (page 2 and Figures 2,3.) I am confused with the choice of $\tau = 1$ s. Previously you write that $\tau \gg TC$, but at the same time you consider $TD = 0.5$ s.

We follow here the practice frequently used in the literature on atomic clocks: For a given clock the Allan deviation $\sigma_y(\tau)$ is determined for long times $\tau \gg 1$ s where $\sigma_y(\tau) = C/\sqrt{\tau}$. The factor C is then used as a figure of merit for comparing different clocks. In the sense of an extrapolation from $\tau \gg 1$ s back to 1 s, the factor C is usually referred to as the Allan deviations at $\tau = 1$ s. (The actual Allan deviation after 1 s of averaging might of course be very different from this value!) In the edited manuscript, we make this clearer by stating (page 2):

“Later on, we will typically refer to Allan deviations at $\tau = 1$ s only. What is meant by this is that we look for the pre-factor to the asymptotic $\sigma_y(\tau) \propto 1/\sqrt{\tau}$ scaling, found e.g. by extrapolating the Allan deviation from a regime with $\tau \gg T_C$ back to $\tau = 1$ s. Even though the actual stability of the clock at $\tau=1$ s may be different, e.g. due to the transient response of the feedback loop, this quantity still provides us with a useful measure to compare the long term stability of different clocks without limitations based on their specific mode of operation.”

7. (page 5) I think that the comparison to [34] is misleading. [34] only studies frequency estimation. It does not provide any predictions for atomic clocks.

(page 6) We restated our comparison to [34]. The part now reads:

“Frequency estimation using GHZ states, which is limited by QPN and atomic decoherence, was already considered quite some time ago in [34]. Here, we have extended this idea to discuss the stability of optical atomic clocks with squeezed states.”

8. (page 9, ref [68]) I think that [Sastrawan, J., Jones, C., Uys, H., Biercuk, M. J., Akhalwaya, I. (2014). Improving frequency standard performance by optimized measurement feedback. Phys. Rev. E, 94(arXiv: 1407.3902), 022204] also considers how to improve the integrator to cancel a non-markovian noise.

We thank the Reviewer for pointing out this publication to us.

(page 10): We included the reference as a possible way to construct optimized linear predictors, generalizing the approach of [68].

9. (page 8-9) I always wondered how reasonable is the assumption that $v_{LO}(t)$ is a prescribed process running on the background. Clearly, this can't be true for large times, but it might be a good approximation. If I understand your point, then you say that this does not matter because the non-white component of the noise can be eliminated by an integrated feedback.

This is one of the assumptions we make. As the referee points out, the integrators in the servo loop transform the laser noise in sections into a spectrum that is on the time scales relevant for the clock servo dynamics well described by this power law expansion. The assumption on the laser behavior is also experimentally verified over time scales of several thousand seconds, see e.g. Matei et al. 2017.

10. (page 9) I do not understand what lower bounds in [67] can be overcome with better servo. In particular, the formula (28) has the same form as bounds in [67]. (more below).

Indeed, the bounds presented in [67] are not overcome with a better servo in general. We referred this statement to the case of random walk noise of the local oscillator as mentioned in the sentence previous to the statement. We have rephrased this part in the revised manuscript to clarify. See also the longer discussion at the end of the personal note for a more detailed showcase how the stability limit of [67] is overcome.

11. (page 9) I like the double-integrator analysis, but at the same time, I am not fully convinced by the results. My main confusion comes from taking the limit $\omega T/g^2 \ll 1$. Is it obvious that this won't interfere with the assumption $g^2 \ll g$? I am worried that if both terms are considered together, then the limit in which the double integrator suppresses the non-white noise is not compatible with g being the dominant term in the feedback. Have you tried a numerical simulation of (15), (16) for a non-markovian noise?

The limit $\omega T/g^2 \ll 1$ will be reached eventually for low enough Fourier frequencies ω , i.e. after sufficiently many cycles of clock operation. Since in the end we are interested in the long-term stability at $\tau \gg T_C$ this is perfectly fine to consider. Thus, in our case the inequality rather states at which point the final stability is reached for given values of T and g^2 . We can thus choose g^2 and g having $g^2 \ll g$ without any restriction by the inequality.

(page 11): We changed the description when going from what is now equation (29) to equation (30) and (31), as well as the discussion of the results presented in (30) and (31).

The simulation of (15) and (16) (which are (19) and (20) in the revised version) with non-Markovian noise for the local oscillator is precisely what we did to obtain the numerical results presented in the main text (symbols in Figures 2, 3 and 5). Actually,

our numerical simulation includes not only the non-Markovian noise but also the non-linearity in the phase estimation as well as the double integrator. Markovian noise is considered only within the analytic models.

(page 11): We included a paragraph specifying the relation between the numerical simulations, as seen in the figures, and the stochastic difference equations presented in the methods.

Personal note (Nature Communications supports signed referee reports): I am confused with the claim that the double integrator can overcome the lower bounds in my work [67]. Specifically, I consider a model with no correlations in different measurement cycles. (I do consider correlations within each cycle, but that should be a minor difference compared to your white noise model (26)). Also the main result of my work is that the clock time variance (for large τ proportional to the Allan variance) is bounded by an expression that is a sum of two factors. The first factor is (proportional to) the inverse of Fisher information F^{-1} , and the second is a non-universal noise dependent factor (that does not depend on N). This looks to me exactly like your expression (29). Maybe I am missing some details; maybe your universal term is better than F^{-1} or the noise term is better than my bound but given that the models are essentially the same this seems unlikely to me.

Of course, there are lots of small differences. I consider only $TD = 0$, I call CLT the Dick effect, I consider discrete time equations etc. I consider an arbitrary measurement and feedback protocol.

I should add that I was very confused about the choice that vLO is uncorrelated between different measurement cycles. At the time that I was working on [67], I asked several people if the model with correlated or uncorrelated noise is better, but did not get any clear answers. In the end, I chose uncorrelated noise just because it was more appealing mathematically. That is, partly, why I like the idea that this can be justified with an integrated servo. Also, I want to add that, apart from giving universal benchmarks, an important part for me was to convince the quantum parameter estimation/ quantum information community that studying atomic clocks with models that do not include the local oscillator noise is missing the point.

Sincerely, Martin Fraas

This is a detailed answer to the personal note of Martin Fraas: At first we remark that the comment we wanted to make was rather generic, saying that in some cases the class of applied feedback has to be extended. To compare to your results, we looked not at the full problem as considered in our manuscript but at a simplified case. We considered only uncorrelated atoms, no squeezing, and zero dead time to be able to compare to your results. As we stated above, the additional structure of the servo is only a problem for strongly correlated noise. Specifically, for the noise types considered in our manuscript, only random walk noise necessarily needs the double integrator to fully cancel the laser noise. This is one of the results from our study, see (29) and (30)

in the revised manuscript, which show that only a non-zero g_2 will fully reduce the laser noise components with a spectral density $S_{LO} \propto 1/\omega^2$. With $g_2=0$ one finds that the overall stability is not only limited by the atomic measurement noise (QPN) but also by a second white noise term which stems from uncorrected laser noise. This will be what your limit (35) in [67] reproduces. We first observed this when we tried to use your general limit (35) in [67] to bound our numerically simulated stabilities. At first we restated your bound (35) in [67] to our stability measure, which is a particular re-scaled Allan deviation expressed in terms of the coherence time Z . This gave $\sigma_y(\tau) \sqrt{2\pi\nu_0} \sqrt{Z\tau} = \sqrt{Z/T_R} * \sqrt{1/N + 3*(Z/T_R)^3}$ as a lower limit to the stability, where we set the gain $g=1-\zeta=1$ and used that the Fisher information is $1/N$ for uncorrelated atoms (explicitly $F_T = T^2/N$ in Ref [67]). When comparing to our numerical simulations with $g_2=0$ this gave an excellent match until the onset of fringe-hops, which is not included in your theory. See Fig.1a for this. However, when we compared to the simulations with $g_2 \neq 0$ we found significantly improved stability compared to your limit, see Fig.1b. The reason for this difference is again the residual white noise component that remains with a simple single-stage feedback. In addition, we note that atomic clocks will always run with a double integrator to also minimize lock errors due to laser drifts.

Figure 1: Comparison of stability limits. **a** Without applying a double integrator in the numerical simulations (symbols), the stability is bound by the stability limit as presented in [67]. **b** When including the double integrator, the stability follows our stochastic differential equation model (dotted lines) and significantly reduces below the bound presented in [67]. Fringe-hops, as observed in the numerics, are not treated in the model of Ref. [67].

Reviewer #2:

The authors propose a theoretical study of the use of spin-squeezing for optical clocks. The main point of the paper is that already for a moderate number of

atoms, the instability is not limited by the QPN (for an optimal timing of the clock sequence), but rather by the technical noise from the ultra-stable laser sources, even taking into account a significant improvement of these sources.

It is indeed no mystery that taking advantage of spin-squeezing would require a significant work on the laser systems, and I congratulate the authors for quantitatively working this out. The paper is very well written and contain enough details to be usefully reproduced. A significant aspect of the paper is the inclusion in the noise model of the phase diffusion process during the probing of the atoms, which is critical to the limits the authors established.

We thank the referee for his/her positive feedback. We are pleased to hear that the referee appreciates the importance of a quantitative assessment of the limits imposed by laser noise on atomic clocks. We hope that the referee will agree with us that knowledge of these limitations is crucial in determining whether or not a given setup for an optical atomic clock can be improved by spin squeezing.

My main concern comes from the universality of the results presented in the paper, that, in my opinion the authors emphasize too much on, starting from the abstract of the paper. There are several interrogations protocols that have been specifically designed to mitigate the Dick effect and the phase diffusion process (Dead time free clocks, weak measurement, possibly with a combination of several atomic ensembles...) and that the authors barely mention in the sentence:

"The limitation could be avoided with schemes achieving dead-time-free interrogation or overcoming laser phase noise [48–54]. The potential gain from entanglement should then be assessed by an appropriate analysis, incorporating the tradeoffs discussed here."

With a bit of exaggeration, I have the feeling that the authors tautologically claim that if one does not take measures to deal with the clock laser noise, then the clock laser noise is limiting.

We think that this last sentence does not present a fair assessment of our results. The argument that the expert develops here in deliberate exaggeration can be applied equally to the treatment of any other noise process: If one does not take measures to deal with noise XX, then XX noise is limiting the clock. This exaggeration makes everything seem trivial.

It is one of the main points of our article that in the context of optical clocks laser noise cannot simply be dismissed as a technicality, as is strongly supported by Referee #1. It is true that there are schemes and ideas for reducing the effects of laser noise in optical clocks, and we admit that our reference to these concepts was too brief and sweeping. In the revised version, we have expanded the discussion of these references to better reflect the possibilities for combating the laser noise limits identified in our work (see changes in the conclusion on page 6). However, we emphasize that all of these ideas and concepts including squeezing have yet to be demonstrated, and their integration and use in real clocks is an outstanding challenge. We insist that our focus on the standard architecture of currently operating atomic clocks - based on single atomic ensembles and Ramsey interrogation using clock lasers with world record stability - is

very well motivated, and highly relevant. Gaining a quantitative understanding of the limits resulting from the interplay of the most dominant noise processes is a non-trivial theoretical challenge that we have addressed in this study. In contrast to the distorting account of the referee, the result does precisely not correspond to the statement that the limiting factor is always laser noise: Sometimes it is laser noise, which is limiting an optical clock, and sometimes other noise sources (such as projection noise) are dominant. In order to make an informed decision as to which measure (reducing projection noise or laser noise) can actually improve the stability of an atomic clock, the limiting factors must be adequately quantified. This is what we provide in our work for the standard architecture of currently operating atomic clocks.

We accept the referee's criticism regarding the too brief description of the methods for overcoming laser noise. In the revision of our article this shortcoming has been corrected. We also try to make our statements not sound more general than they are by clearly stating in several places to which type of atomic clock we refer. (This information was already in the abstract before). We hope that these measures adequately address the criticism of the referee. More specifically, we applied the following changes:

i) (page 6) In the conclusion we changed "*The limitations described here may be overcome with more sophisticated clock architectures.*" to "*The limitations described here, valid for single ensemble clocks with cyclic Ramsey interrogation and dead time, may be overcome with more sophisticated clock architectures.*" in order to not claim more universality of our results than proven. Furthermore, we extended the existing discussion of specifically designed protocols to mitigate either the CTL or remove dead time in the conclusion. But we want to stress that a full treatment of all these approaches in the way we present it in our original work is far beyond the scope of the paper.

ii) (page 6): We added the statement: "*The model we developed allows a comprehensive and quantitative investigation in which parameter regimes laser noise is not the most stringent limitation, so squeezing can improve the stability, and in which cases laser noise is dominant and needs to be overcome by other means before squeezing provides an advantage.*" to address the possible interpretation of our claim regarding laser noise. But we would like to still stress that laser noise is an essential part of optical atomic clocks and has to be treated in detail. See also the related statements of reviewer #1.

Another important aspect that I think is missing (it is only dealt with in the supplementary materials on a specific case) is the duration of the Ramsey interrogation that corresponds to the minimal Allan deviation. There are several practical limitations to this duration that may be difficult to overcome (coherence losses by collisions, photon scattering, natural linewidth...), that may very well alter the maximal number of atoms for which SSS are useful. All in all, I think it would be hard to operate an optical lattice clock with a Ramsey dark time large than typically 1 min. I think it would be of interest that this aspect is incorporated in the graphs of figure 2.

The question of an additionally restricted interrogation time is an interesting objection by the reviewer. Given that there are many different reasons for a maximum Ramsey

duration, as the reviewer herself/himself points out, we believe this issue should best be examined on a case-by-case basis according to the physical origin and the respective effects on the clock stability.

In order not to go into such details, we restrain ourselves first to a purely qualitative discussion about the implications of a maximum Ramsey duration on our results and then illustrate the quantitative change on one concrete example.

(page 5): We added: “*Finally, the results presented above may be altered if there exists some additional process which places an upper bound $T_{\{max\}}$ to the Ramsey time. This could occur due to coherence losses from collisions, photon scattering, a limited natural lifetime or others. Of course, in the case $T_{R^*} \leq T_{\{max\}}$, where $T_{\{max\}}$ is larger than the optimal interrogation time T_{R^*} identified above, our results are unchanged. T_{R^*} is on the order of a few seconds for cL , see supplementary information.*

When $T_{R^} \geq T_{\{max\}}$ one can define the new critical particle number $\tilde{N}_{\{min\}}(T_{\{max\}}) = \min_N \{ \sigma_{\{QPN\}}(N, T_{\{max\}}) \leq \sigma_{\{Dick\}}(T_{\{max\}}) \}$. For example, at $T_D = 0.1$ s and assuming the laser cL , we find that the critical particle number changes from $N_{\{min\}} = 1244$ to $\tilde{N}_{\{min\}}(T_{\{max\}}=0.1$ s) = 3504, $\tilde{N}_{\{min\}}(T_{\{max\}}=1$ s) = 1475 and remains unchanged if $T_{\{max\}} > 2.6$ s.”*

So in conclusion, I think that the paper is extremely useful for the community because of the high quality of the model the authors developed and explained. It is fine if the authors apply this model in a particular interrogation sequence (that is currently typically employed in optical clocks), but I think the main message that the paper conveys (basically that the usefulness of spin squeezing is essentially limited to optical clocks with a rather small number of atom) overcomes what is actually proven in the paper. I think the authors should more quantitatively consider this before publication, or at least make it more clear. The possible restricted universality of the results presented by the authors could limit the interest of the paper to a broad scope journal like nature comm.

We thank the reviewer for his/her positive assessment of our work and we hope to have addressed the concerns with the changes mentioned above. Regarding the choice of journal, we do believe that Nature Communications is a good fit for our manuscript. In Nature Communications “guide to authors” it is explicitly stated that “Papers published by the journal represent important advances of significance to specialists within each field.” and regarding articles Nature Communications states that “[...] an Article is a novel and important research study of high quality and of interest to that specific research community”.

All reviewers have expressed that our work is of high quality, reviewer #2 and reviewer #3 both expressed that our work is of significant relevance to the atomic clock community and one can see from the comments of reviewer #1 that our work may also be of high interest to the broader field of research, including e.g. the quantum parameter

estimation/quantum information communities. Thus, we believe that our work fits nicely within the scope of Nature Communications.

Reviewer #3:

The authors analyze the stability of optical atomic clocks using spin squeezed states in the presence of realistic laser phase noise and dead time in the clock interrogation cycle. Analytic calculations for the asymptotic stability are derived and confirmed with numerical simulations. The results are presented in various parameter space regions including low atom-number and high atom-number as well as better or worse laser phase noise. Based on their analysis, the authors conclude that with current clock laser technology, squeezing only helps clocks with small (order 10) atom number, but for clocks using improved future clock lasers squeezing can help even for large atom numbers.

This analysis will certainly be of interest to the atomic clock and quantum metrology communities, but it's not obvious to me that it is of general enough interest for the audience of Nature Communications. The introduction requires quite a bit of physics background knowledge to be fully understood. The qualitative description filling the first 4 pages of the manuscript is largely already known and various regimes have been described elsewhere, although this is perhaps the most clear and complete description I am aware of. The last page of main text and the methods section contain new analytic calculations, which are quite interesting but may be difficult for a broad audience to follow. Perhaps Nature Physics or PRL would be a better fit?

Here we refer to the reasoning we gave when addressing reviewer #2. In short, we do believe Nature Communications is a good fit for our work as it features high quality research within specific disciplines and is raising interest within related research directions.

Specific comments follow, in no particular order:

1. Page 5 - This abbreviated description of the SDE analysis is difficult for me to follow, and I consider myself an expert in this field. I think that people in fields less quantitative than physics will have almost no hope of understanding. It might be better if the discussion of the SDE analysis is entirely moved to the methods (although then the main text would not contain any description of the novel part of this work), or if the entire methods section is incorporated at the end of the main text in a longer article format journal. Some symbols are undefined or use un-descriptive symbols, further compounding the difficulty of understanding (i.e., What does σ_{eff} stand for in σ_{eff} ? Effective doesn't really make sense in this context. Furthermore, it's ambiguous whether σ_{eff} is the total clock instability or just the contribution from CTL, and it's not explicitly defined).

We are aware that this last section of the main text is difficult to follow due to its brevity. However, we have knowingly chosen to focus the main text mostly on a discussion of

our results making the main points as clear as possible. And we are pleased that the reviewer also calls this "perhaps the most clear and complete description I am aware of". We believe that interested readers will also look closely at the Methods section to find the details of our models without necessarily including them in a longer version of our manuscript.

Regarding the un-descriptive symbols we revised this section in the following way:

(page 5-6) We renamed the variance as V_{m+d} where the subscript is a shorthand for "measurement with diffusion". In the revised manuscript we clarify the meaning of this variance by stating: "[...] where V_{m+d} is the variance of measurement outcomes when the dynamics is affected by laser phase diffusion. As this is a combined description of measurement noise, leading to QPN, and phase diffusion, leading to the CTL, one can infer both contributions from V_{m+d} as we show below."

2. Figure 2 (especially the 10 atom case) suggests to me that CTL and fringe hops contribute at a similar level. I say this because the data points for the simulated clock stop (due to fringe hops according to the text) somewhat before T_R^* and the best stability is ~20% above the curve adding QPN, CTL, and Dick effect. This is supported by Figure 4 in the methods which shows that $T_{FH} < T_R^*$ for 10 ions and 0.5 s dead time. However, the main text (right column of page 2) says that CTL is a "more stringent limit" than fringe hopping. Would it be more accurate to say that CTL and fringe hopping contribute at a similar level?

This is correct. The parameters used for Figure 2 are on the border of what we would describe as the "good atomic clock" regime in our manuscript. In this case fringe-hops and CTL contribute at a similar level. The quote the reviewer refers to was not meant explicitly with respect to Figure 2, but rather to "the regime of a good atomic clock (long laser coherence time and small dead time)" as mentioned in the first part of the quoted sentence. But we agree that it would be best to relax our claim at this point.

To this end the sentence in question (page 2) now reads "*We find that in the regime of a good atomic clock (long laser coherence time and small dead time) fringe-hops and the CTL contribute either at a similar level or the diffusive process σ_{CTL} constitutes the more stringent limitation for the Ramsey interrogation, so that we concentrate the discussion on the latter effect.*"

We hope the reader considers Figure 4 for a more quantitative analysis of this statement.

3. Figures 2 & 3 - The transition frequency of the stimulated clock should be specified, since this is necessary to reproduce the plots.

(page 5) We removed the statement: "*When necessary we use the frequency $\nu_0 \approx 429.228$ THz of ^{87}Sr for calculations.*"

Instead we now write "*We use the transition frequency $\nu_0 \approx 429.228$ THz of ^{87}Sr for calculations.*" in the caption of both Figure 2 and Figure 3.

4. Table 1 - Similarly, the laser frequency should be specified, since the coherence time depends on it.

We agree and now provide the value of ν_0 in the caption of Table 1:

“Then with $\nu_0 = 429.228$ THz the coherence time Z is determined as defined in the main text.”

5. Above equation 13 - It seems rather disjointed to refer the reader to the supplemental information for the description of the statistics, especially since the discussion in the supplement is only one paragraph long. I'd just move section I of the supplement to the location in the methods where it is referenced.

We removed section I of the supplementary information and added the relevant information above equation 13 (see highlighted changes).

6. Equation 13 - "dt" is not defined.

(page 10): We included the definition of dt by adding the sentences:

“The differential time increment $dt = T$ corresponds to the Ramsey duration as we are interested in studying the phase evolution over the course of many interrogation cycles. [...]. We also did not cancel the factors T in the first term on the right-hand side to highlight the correspondence to the continuous stochastic differential equation.”

7. Below equation 27 - Although V_{ϕ} is defined at the end of the main text, it would be useful to remind the reader what it means here.

(page 12 - below Eq.27 (Eq.33 in the revised version)): To remind the reader about the meaning of V_{ϕ} we state:

“Here the variance V_{ϕ} quantifies again the width of the phase distribution prior to each measurement. The greater the correlations in the laser noise, the faster the width V_{ϕ} of the phase distribution increases with the Ramsey time.”

8. Figure 4 - Where do the kinks in the boundary between the red and blue regions come from? I would have expected this boundary to be smooth.

(Figure 4) The kinks are due to finite sampling of the two-dimensional $T_D - Z$ plane (we used a 21x21 grid). With a finer grid of data points the boundary is expected to be smoothed out.

9. Page 12 - Why is a different interval used for flicker vs random walk? In both cases, the phase estimate is wrong if the actual phase is outside $(-\pi/2, \pi/2)$.

It is indeed true that a phase outside $(-\pi/2, \pi/2)$ can no longer be estimated correctly. Although the applied feedback for phases between $\pi/2$ and π (and analogously between $-\pi/2$ and $-\pi$) gives insufficient corrections, the stabilization in this case still remains on the correct fringe. This is because the servo correction is still applied in the

correct direction, back to phase=0. For phases with $|\phi| > \pi$, not only is the feedback insufficient, but the feedback also corrects in the wrong direction. Naively speaking, one would assume that in this case the stabilization is definitely transferred to an adjacent fringe and strictly speaking, the escape time from the interval $(-\pi, \pi)$ should always be examined. For flicker noise we indeed found good agreement with the full simulations of the atomic clock when considering this interval.

For random-walk noise, however, we found a slightly better agreement with the simulations when we examine the escape time from $(-\pi/2, \pi/2)$. We suspect that due to the strong temporal correlations of this noise type it appears the phase can no longer be sufficiently corrected from this point on and fringe-hops occur.

(page 13) We revised our manuscript in the following way:

The sentence

“A maximum Ramsey time $T_{\{FH\}}$ without fringe-hops can be specified by requesting that the stabilized phase should not leave the interval $(-\pi, \pi)$ for flicker frequency noise – where it is corrected back to the original reference point – or $(-\pi/2, \pi/2)$ for the more strongly correlated random walk noise, within the simulated $\sim 10^6$ cycles of clock operation.”

was replaced by

“A maximum Ramsey time $T_{\{FH\}}$ without fringe-hops can be specified by requesting that the stabilized phase should not leave the interval $(-\pi, \pi)$, where it is corrected back to the original reference point $\phi = 0$, within the simulated $\sim 10^6$ cycles of clock operation. [...]

In the case of flicker frequency noise this led to a good agreement with the onset of fringe-hops observed in numerical simulations. For random walk noise we achieved slightly better agreements when assuming the interval $(-\pi/2, \pi/2)$ for the calculation of the mean first passage time. We found that this stronger requirement is more applicable here due to the increased temporal correlations which already cause fringe-hops in a regime where the feedback, though insufficient, is not on paper stabilizing to a different fringe.”

10. Figure 5 - The x and y scales use very small font; it is difficult to read even on a large monitor.

(Figure 4 and 5) We re-worked Figure 5 to enhance the x and y scales as well as the ticks. In doing so we noticed the same issue with Figure 4 so we re-worked the scales and ticks there as well.

Reviewers' Comments:

Reviewer #1:

Remarks to the Author:

Dear Editor,

In the response letter, the authors answered all my concerns. They also implemented all the relevant changes in the new version of the manuscript. I want to, in particular, thank the authors for their very detailed explanation of the differences with respect to [67]. The model in [67] indeed does not directly apply to a situation with a strongly correlated noise. The new text makes it clear that this is the shortcoming that they have in mind.

My recommendation remains unchanged, I enthusiastically recommend publication of the article.

I also want to make two additional comments about the discussion in the response letter.

1. I do think that the work is interesting to quantum estimation community. I can think of handful of researches (including myself) that might in the future try to analyze equations (21), (22) from the information theory point of view.

2. It is fascinating to see the range of views on what are the important aspects. From a purely theoretical point of view, it is very important to have a minimal model for atomic clocks. This means a model that includes all relevant phenomena but no technical noise. I agree with the authors that (unlike many other sources of noise) the laser noise have to be included in the minimal model. Without laser noise, the optimal feedback is no feedback, and the model will not retain any phenomena associated with atomic clocks.

Reviewer #2:

Remarks to the Author:

In response to the authors' answer to the first review, I would like to raise to following points:

* Concerning my first comment about the generality of the model: I am sorry if I have been unclear, or a bit too sarcastic, but my point was certainly not to argue that the clock laser noise is not important. On the contrary, I completely agree with the authors and with the first reviewer that duly taking into consideration the clock laser noise - as is rigorously done in this paper - is critical. But so does the community: this is why all these astute protocols (dead time free clocks, synchronous interrogations...) have been designed and are actually being implemented, in order to escape the "usual cyclic Ramsey interrogation of single atomic ensembles with dead time" that is mentioned in the abstract, and that is the framework of this paper. For sure, SSS can only help if the clock stability is largely dominated by the quantum projection noise, and efforts to reach this goal are a prominent research axis. In my view, downplaying the relevance of these efforts goes against the wish of the authors to point out that the "clock laser noise cannot simply be dismissed as a technicality".

In this respect, I think the authors significantly improved the manuscript with the discussion they added in the conclusion (page 6). However, I think it would have been more appropriate to put the scope of their model into its context at the very beginning, starting from the abstract and the introduction.

Also, I'd like to emphasize that beyond the scientific facts they present (which are perfectly valid here), papers convey a more qualitative message, esp. for papers published in journals like Nat. Comm. As the paper is written now, I can hear a message saying "In this paper, we show that SSS may be useless for optical lattice clocks". I had the opportunity to hear colleagues discussing this paper (arxiv version), and I feel this impression is shared. However, I would say that, beyond the

loophole discussed above, this message is not such a revolution: it is quite obvious that, in order to benefit from SSS, the clock stability has to be largely dominated by the QPN, which is still far from being the case for current optical lattice clocks, especially when pushing T_R as far as the laser coherence permits. And this conclusion, that could be drawn from a very naive calculation of the Dick effect limited stability, is not changed much by the inclusion of a stochastic model for the CTL.

In summary, while I praise the quality of the paper for its scientific rigor, the in-depth view it offers, and the completeness of the explanation (esp. in the methods), I still think that it leads to an unsurprising conclusion in a limited framework. This is why it may fail to be in accordance with the editorial line of Nature Communications.

* Concerning the possibility that the Ramsey dark time can be limited to T_{\max} : I agree with the authors that this is a difficult topic to discuss given the multiple reasons for which this time can be limited. What I had in mind in my initial comment was to add a plot that shows T_R^* for all the configurations considered in Fig 3, so that the reader can compare this T_R^* to her favorite value of T_{\max} . The authors preferred to give the value of T_R^* in the main text for a specific configuration (clock laser cL). Given that this value of T_R^* is reasonably short, I am perfectly fine with this choice, provided that T_R^* does not differ dramatically for pL1 or pL2.

Reviewer #3:

Remarks to the Author:

The authors have addressed all of my concerns and suggestions in this revised draft. Furthermore, the author's reply to referee letter has changed my opinion about the suitability of this work for Nature Communications - although the results are somewhat narrowly focused, I do agree that it would be rather valuable to expose readers in other disciplines to these results, as a way to attract new people and ideas to this line of research and vice versa. I recommend publication in Nature Communications as is.

Dear editor,

we are again grateful to all reviewers for the time and efforts they invested in evaluating the quality of our revised manuscript. We are pleased to so see that our revision addressed all concerns that the reviewers brought forth and that, after final revision, the manuscript has been accepted for publication in Nature Communications.

Based on the reviewers' comments we see no further point which requires changes to the manuscript.

For all authors,

Marius Schulte

Reviewer #1 (Remarks to the Author):

Dear Editor,

In the response letter, the authors answered all my concerns. They also implemented all the relevant changes in the new version of the manuscript. I want to, in particular, thank the authors for their very detailed explanation of the differences with respect to [67]. The model in [67] indeed does not directly apply to a situation with a strongly correlated noise. The new text makes it clear that this is the shortcoming that they have in mind.

My recommendation remains unchanged, I enthusiastically recommend publication of the article.

I also want to make two additional comments about the discussion in the response letter.

1. I do think that the work is interesting to quantum estimation community. I can think of handful of researches (including myself) that might in the future try to analyze equations (21), (22) from the information theory point of view.

2. It is fascinating to see the range of views on what are the important aspects. From a purely theoretical point of view, it is very important to have a minimal model for atomic clocks. This means a model that includes all relevant phenomena but no technical noise. I agree with the authors that (unlike many other sources of noise) the laser noise have to be included in the minimal model. Without laser noise, the optimal feedback is no feedback, and the model will not retain any phenomena associated with atomic clocks.

We thank the referee for his positive feedback.

Reviewer #2 (Remarks to the Author):

In response to the authors' answer to the first review, I would like to raise to following points:

* Concerning my first comment about the generality of the model: I am sorry if I have been unclear, or a bit too sarcastic, but my point was certainly not to argue that the clock laser noise is not important. On the contrary, I completely agree with the authors and with the first reviewer that duly taking into consideration the clock laser noise - as is rigorously done in this paper - is critical. But so does the community: this is why all these astute protocols (dead time free clocks, synchronous interrogations...) have been designed and are actually being implemented, in order to escape the "usual cyclic Ramsey interrogation of single atomic ensembles with dead time" that is mentioned in the abstract, and that is the framework of this paper. For sure, SSS can only help if the clock stability is largely dominated by the quantum projection noise, and efforts to reach this goal are a prominent research axis. In my view, downplaying the relevance of these efforts goes against the wish of the authors to point out that the "clock laser noise cannot simply be dismissed as a technicality".

In this respect, I think the authors significantly improved the manuscript with the discussion they added in the conclusion (page 6). However, I think it would have been more appropriate to put the scope of their model into its context at the very beginning, starting from the abstract and the introduction.

Also, I'd like to emphasize that beyond the scientific facts they present (which are perfectly valid here), papers convey a more qualitative message, esp. for papers published in journals like Nat. Comm. As the paper is written now, I can hear a message saying "In this paper, we show that SSS may be useless for optical lattice clocks". I had the opportunity to hear colleagues discussing this paper (arxiv version), and I feel this impression is shared. However, I would say that, beyond the loophole discussed above, this message is not such a revolution: it is quite obvious that, in order to benefit from SSS, the clock stability has to be largely dominated by the QPN, which is still far from being the case for current optical lattice clocks, especially when pushing T_R as far as the laser coherence permits. And this conclusion, that could be drawn from a very naive calculation of the Dick effect limited stability, is not changed much by the inclusion of a stochastic model for the CTL.

In summary, while I praise the quality of the paper for its scientific rigor, the in-depth view it offers, and the completeness of the explanation (esp. in the methods), I still think that it leads to an unsurprising conclusion in a limited framework. This is why it may fail to be in accordance with the editorial line of Nature Communications.

While we disagree with the final conclusion of the reviewer, based on his/her personal concerns, we nevertheless thank him/her for the positive assessment on the scientific quality of our work.

* Concerning the possibility that the Ramsey dark time can be limited to T_{\max} : I agree with the authors that this is a difficult topic to discuss given the multiple reasons for which this time can be limited. What I had in mind in my initial comment was to add a plot that shows T_R^* for all the configurations considered in Fig 3, so that the reader can compare this T_R^* to her favorite value of T_{\max} . The authors preferred to give the value of T_R^* in the main text for a specific configuration (clock laser cL). Given that this value of T_R^* is reasonably short, I am perfectly fine with this choice, provided that T_R^* does not differ dramatically for pL1 or pL2.

The values of T_R^* differ by a factor of 5 and 16 only, for pL1 and pL2 respectively.

Reviewer #3 (Remarks to the Author):

The authors have addressed all of my concerns and suggestions in this revised draft. Furthermore, the author's reply to referee letter has changed my opinion about the suitability of this work for Nature Communications - although the results are somewhat narrowly focused, I do agree that it would be rather valuable to expose readers in other disciplines to these results, as a way to attract new people and ideas to this line of research and vice versa. I recommend publication in Nature Communications as is.

We are glad to see the reviewer acknowledges the relevance of our work and thank him/her for the recommendation to publish in Nature Communications.